# A unified framework for measuring selection on cellular lineages and traits

Shunpei Yamauchi[1], Takashi Nozoe[1], Reiko Okura[1], Edo Kussell[2,3], Yuichi Wakamoto[1,4,5]*

[1]Department of Basic Science, Graduate School of Arts and Sciences, The University of Tokyo, Tokyo, Japan; [2]Department of Biology, New York University, New York, United States; [3]Department of Physics, New York University, New York, United States; [4]Research Center for Complex Systems Biology, The University of Tokyo, Tokyo, Japan; [5]Universal Biology Institute, The University of Tokyo, Tokyo, Japan

**Abstract** Intracellular states probed by gene expression profiles and metabolic activities are intrinsically noisy, causing phenotypic variations among cellular lineages. Understanding the adaptive and evolutionary roles of such variations requires clarifying their linkage to population growth rates. Extending a cell lineage statistics framework, here we show that a population's growth rate can be expanded by the cumulants of a fitness landscape that characterize how fitness distributes in a population. The expansion enables quantifying the contribution of each cumulant, such as variance and skewness, to population growth. We introduce a function that contains all the essential information of cell lineage statistics, including mean lineage fitness and selection strength. We reveal a relation between fitness heterogeneity and population growth rate response to perturbation. We apply the framework to experimental cell lineage data from bacteria to mammalian cells, revealing distinct levels of growth rate gain from fitness heterogeneity across environments and organisms. Furthermore, third or higher order cumulants' contributions are negligible under constant growth conditions but could be significant in regrowing processes from growth-arrested conditions. We identify cellular populations in which selection leads to an increase of fitness variance among lineages in retrospective statistics compared to chronological statistics. The framework assumes no particular growth models or environmental conditions, and is thus applicable to various biological phenomena for which phenotypic heterogeneity and cellular proliferation are important.

*For correspondence:
cwaka@mail.ecc.u-tokyo.ac.jp

Competing interest: The authors declare that no competing interests exist.

## Editor's evaluation

This manuscript presents a general statistical framework to infer selection on a quantitative trait, based on measurements of the values of this trait along related cell lineages. The manuscript provides both a detailed explanation of the mathematical underpinnings of the method and an illustration of its application to existing and new cell lineage datasets. This is a general framework and is not tailored to particular growth models or environmental conditions, making it applicable to broad examples of exponentially growing populations.

## Introduction

Growth rates of cellular populations are physiological quantities directly linked to the fitness of cellular organisms. To understand the roles of biological processes and reactions within cells, including modulation of gene expression and metabolic states, one must characterize how they are eventually channeled into an increase or maintenance of population growth rates.

As documented by many single-cell studies, phenotypic states of individual cells in cellular populations are heterogeneous and often correlate with fitness variations among cellular lineages (*Balázsi et al., 2011*; *Elowitz et al., 2002*; *Kelly and Rahn, 1932*; *Powell, 1956*; *Wakamoto et al., 2005*; *Wang et al., 2010*; *Cerulus et al., 2016*; *Susman et al., 2018*). Fitness heterogeneity within a population causes a statistical bias on ancestral cells' contributions to the number of descendants, which is broadly referred to as 'selection' (*Leibler and Kussell, 2010*). Such bias from growth heterogeneity makes the relations between cellular lineages and populations nontrivial. For example, an intriguing consequence of intra-population selection is a growth rate gain, a phenomenon that cell population's growth rate becomes greater than the mean division rate of isolated single-cell lineages (*Powell, 1956*; *Hashimoto et al., 2016*; *Rochman et al., 2018*). Recent progress of single-cell measurements has enabled high-throughput acquisitions of cellular lineage trees and historical dynamics in each lineage (*Stewart et al., 2005*; *Wang et al., 2010*; *Hashimoto et al., 2016*). However, establishing the theory and method of cellular lineage statistics to quantify fitness differences among different phenotypic states and intrapopulation selection is still in progress (*Nozoe et al., 2017*; *García-García et al., 2019*; *Levien et al., 2020*; *Genthon and Lacoste, 2020*; *Genthon and Lacoste, 2021*).

Growth of cellular populations can be described using the ensemble of individual cells' growth histories (*Leibler and Kussell, 2010*). A theoretical approach that regards a cell lineage (history) as a basic unit of analysis has offered illuminating insights into population dynamics. For example, it has provided the formula for untangling selection from responses (*Leibler and Kussell, 2010*), population response to age-specific changes in mortality and fecundity (*Wakamoto et al., 2012*), fluctuation relations of fitness (*Kobayashi and Sughiyama, 2015*; *Genthon and Lacoste, 2020*), and relations between cell size growth rate and population growth rate (*Thomas, 2007*; *Lin and Amir, 2017*).

Employing this cell history-based formulation of population dynamics, we have previously proposed a method of cellular lineage statistics that allows quantification of fitness landscapes and selection strength for any traits of cellular lineages (*Nozoe et al., 2017*). Here, we extend this statistical framework and show that population growth rates can be expanded by the cumulants that represent various properties of fitness distributions, such as variance and skewness, in a population. We apply the framework to experimental single-cell lineage data of bacteria, yeast, and mammalian cells to quantify their condition-dependent growth heterogeneity and its contribution to population growth rate. We also apply this framework to measuring the fitness landscapes for a growth-regulating sigma factor in *E. coli* and identify the conditions where its continuum and non-genetic expression heterogeneity correlates with lineage fitness in cellular populations.

## Examples of biological questions

Before detailing the theoretical and experimental results, we first present several biological questions for which cell lineage statistics could provide essential insights.

## Growth rate gain

Growth of individual cells is heterogeneous in a cellular population even under constant environmental conditions (*Stewart et al., 2005*; *Wakamoto et al., 2005*; *Wang et al., 2010*; *Hashimoto et al., 2016*). Whether genetic or non-genetic, such growth heterogeneity inevitably enables selection within a cellular population. Growth heterogeneity can increase the rate of a population's growth compared to the mean replication (division) rate of individual cells, known as 'growth rate gain' (*Hashimoto et al., 2016*). Since population growth rate is one of the critical quantities that determine long-term evolutionary success, it is interesting to ask to what extent growth heterogeneity contributes to population growth rate and how the contributions change depending on cellular phenotypes, genotypes (e.g. species), and environmental conditions. Answering this question may uncover strategies of each organism regarding how it exploits inherent stochasticity for population growth.

As we detail below, a measure of selection strength, $S_{\mathrm{KL}}^{(1)}[D]$, can quantify the growth rate gain from growth heterogeneity. Furthermore, we show that one can quantitatively decompose $S_{\mathrm{KL}}^{(1)}[D]$ into the contributions of distinct characteristics of growth heterogeneity, such as variance and skewness of fitness distributions. In this study, we apply the cell lineage statistics framework to single-cell lineage data and unravel how the growth rate gain changes across environments and organisms.

## Selection in changing environments

When a population of cells faces environmental changes, response of individual cells can be uniform and heterogeneous (*Lambert et al., 2014*; *Julou et al., 2020*). In one scenario, individual cells might respond to an environmental change uniformly and contribute to the future population nearly equally with respect to the number of descendants. In another scenario, only a tiny fraction of the cell population could respond to an environmental change, and the descendants of the responders might dominate the entire future population. In this case, the selection within a population is intense, and the nature of a population's response exclusively depends on these rare cell lineages. Typically, the responses of real cell populations would fall between these two extremes; it is therefore critical to ask how strongly selection occurs within cellular populations in response to environmental changes to understand their response and adaptation strategies.

The framework enables such quantification by evaluating the selection strength $S_{\mathrm{KL}}^{(1)}[D]$ of responding cell populations. Importantly, quantifying the selection strength $S_{\mathrm{KL}}^{(1)}[D]$ requires only the information of division counts in cellular lineages. Hence, the selection strength is measurable even for complex processes where clarifying the transitions of environmental conditions around cells is technically challenging. We indeed analyze cellular populations of *E. coli* regrowing from an early or late stationary phase and characterize distinct levels of selection depending on the duration of stationary phase.

## Correlations between cellular lineage traits and fitness

Since various traits of individual cells, such as expression levels of particular genes (*Elowitz et al., 2002*), are heterogeneous in cellular populations, it is natural to ask how strongly trait heterogeneity correlates with the fitness of individual cell lineages. Quantifying such correlations will allow us to understand which traits are under strong selection and potentially crucial for long-term evolution.

The cell lineage statistics framework quantifies relationships between traits and fitness using fitness landscapes $h(x)$. Additionally, the overall correlation between the heterogeneity of traits and that of fitness can be quantified by the relative selection strength $S_{\mathrm{rel}}[X]$. In this study, we measure $h(x)$ and $S_{\mathrm{rel}}[X]$ for a growth-regulation sigma factor in *E. coli* to unravel whether its continuum expression level heterogeneity is correlated with the fitness heterogeneity of single cell lineages.

Clarifying trait and fitness correlations based on individual-cell-based analyses is difficult when growth and traits fluctuate rapidly over time and when the traits affect growth with delays. In such circumstances, instantaneous correlations between traits and growth might not report their relations correctly. On the other hand, the cell-lineage-based analysis of this framework can take the whole dynamics of traits in cell lineages into account. For example, if we expect that absolute expression levels are important for fitness, the expression level averaged in each cell lineage can be employed as the lineage trait, and its fitness landscape and selection strength are measurable. If large fluctuations affect cell fates and contribute to diversification of cell lineage fitness within a cellular population (*Purvis and Lahav, 2013*), the variances of expression levels can be taken as lineage traits, and one can evaluate their fitness landscape $h(x)$ and relative selection strength $S_{\mathrm{rel}}[X]$. Therefore, the assumption of a cell lineage as a unit of selection can significantly extend the choice of traits, including time-dependent properties, and can provide insights into cellular dynamics that cannot be gained without the lineage-based formulation of fitness and selection.

## Theoretical background

First, we briefly review the analytical framework of cell lineage statistics introduced in *Nozoe et al., 2017*. This framework allows us to quantitatively infer fitness differences associated with distinct states of cellular lineage traits and selection within a growing cell population from empirical single-cell lineage tree data. Time-lapse single-cell measurements provide cellular growth and division information in the form of lineage trees (*Figure 1*, *Stewart et al., 2005*). We regard a lineage $\sigma$ as a cell history traceable back from a descendant cell at the final time point $t = \tau$ (*Figure 1B*). For the case of cellular growth shown in *Figure 1A*, 22 cell lineages exist in the trees.

We assign two types of probability weight to cellular lineages. One is retrospective probability, in which we assign equal weight $P_{\mathrm{rs}}(\sigma) := 1/N_\tau$ to all lineages, where $N_\tau$ is the number of cells at the final time point $t = \tau$. $P_{\mathrm{rs}}(\sigma)$ represents the probability of selecting the history of a cell present at the endpoints of lineage trees. Another is chronological probability, in which we assign the weight

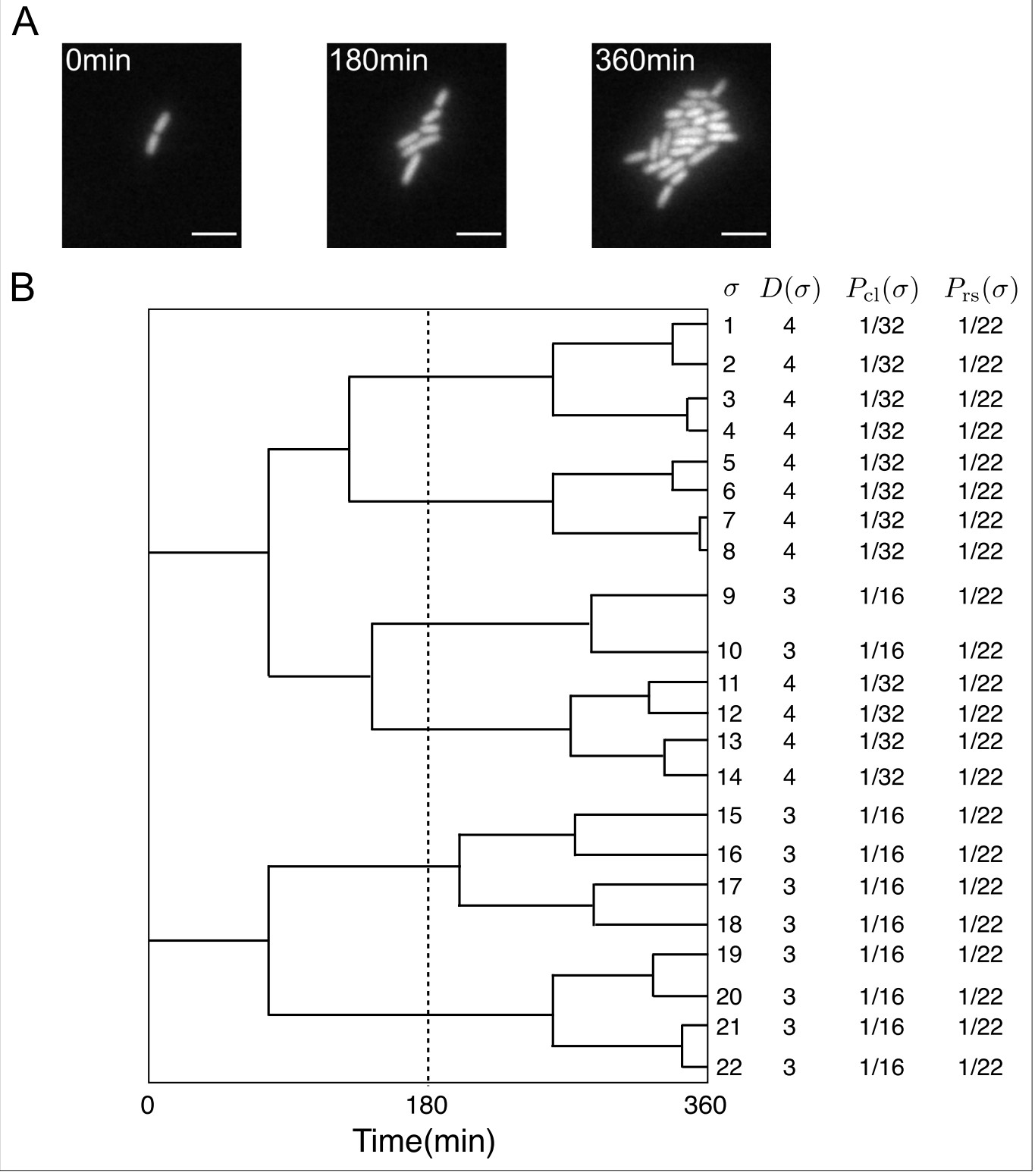

**Figure 1.** Representative single-cell lineage trees. (**A**) Time-lapse images of a growing microcolony of *Escherichia coli* expressing green fluorescent protein (GFP) from plasmids. Scale bars, 5 µm. (**B**) Cellular lineage trees for the microcolony in A. Bifurcations in the trees represent cell divisions. $\sigma$ denotes cell lineage labels. $D(\sigma)$ shows the number of cell divisions in each lineage. $P_{\mathrm{cl}}(\sigma)$ and $P_{\mathrm{rs}}(\sigma)$ are chronological and retrospective probabilities defined in the main text.

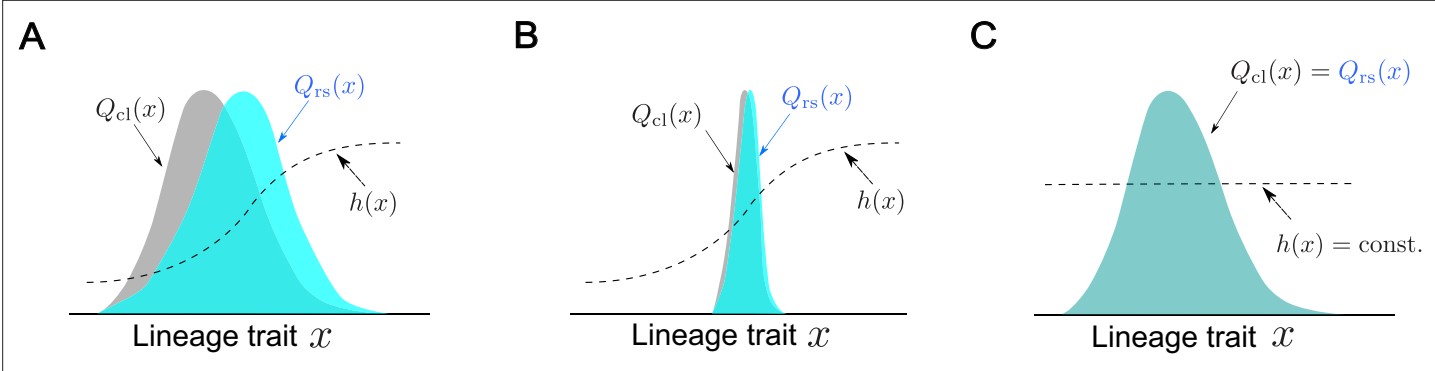

**Figure 2.** Conceptual illustration of the relationships between fitness landscapes, trait distributions, and selection strength. (**A**) Non-uniform fitness landscape and broad trait distribution. The gray distribution represents a chronological distribution of lineage trait $x$; the cyan distribution represents a retrospective distribution of lineage trait $x$; and the black dashed line represents a fitness landscape. Due to the non-uniform fitness landscape and the broad chronological distribution, there is trait fitness heterogeneity for selection to act on. The retrospective distribution therefore shifts significantly from the chronological distribution, and the selection strength is large ($S[X] > 0$). (**B**) Non-uniform fitness landscape and narrow trait distribution. Due to the lack of trait heterogeneity, there is little fitness heterogeneity for selection to act on. The retrospective distribution shifts only slightly from the chronological distribution, and the selection strength is small ($S[X] \approx 0$). (**C**) Uniform fitness landscape. When the fitness landscape is constant ($= \tau\Lambda$) across the lineage trait state $x$, there can be no trait fitness heterogeneity regardless of whether the trait distribution itself is narrow or broad. The selection strength is therefore zero ($S[X] = 0$).

$P_{\mathrm{cl}}(\sigma) := 2^{-D(\sigma)}/N_0$ to the lineages, where $D(\sigma)$ is the number of cell divisions on lineage $\sigma$ and $N_0$ is the initial number of cells at $t = \tau$. $P_{\mathrm{cl}}(\sigma)$ represents the probability of choosing lineage $\sigma$ descending the tree from one of the ancestor cells at $t = 0$ and selecting one branch with the probability 1/2 at every cell division. $P_{\mathrm{rs}}(\sigma)$ and $P_{\mathrm{cl}}(\sigma)$ can be different in general when the number of cell divisions are variable among the cell lineages, as shown in *Figure 1B*.

We define retrospective and chronological probabilities for a *lineage trait* $X$ as $Q_{\mathrm{rs}}(x) := \sum_{\sigma:X(\sigma)=x} P_{\mathrm{rs}}(\sigma)$ and $Q_{\mathrm{cl}}(x) := \sum_{\sigma:X(\sigma)=x} P_{\mathrm{cl}}(\sigma)$, where $X(\sigma)$ is the value of trait $X$ for lineage $\sigma$. Here, we regard any measurable quantity associated with cellular lineages as a lineage trait $X$. For example, time-averaged expression levels and production rates of a drug-resistance protein were analyzed as lineage traits in the experiments of *Nozoe et al., 2017*. Intuitively, $Q_{\mathrm{cl}}(x)$ and $Q_{\mathrm{rs}}(x)$ represent the probabilities of finding the lineage trait value $X = x$ before and after selection, respectively.

Using these retrospective and chronological distributions, we define the *fitness landscape* for lineage trait $X$ as

$$h(x) := \tau\Lambda + \ln \frac{Q_{\mathrm{rs}}(x)}{Q_{\mathrm{cl}}(x)}, \tag{1}$$

where $\Lambda := \frac{1}{\tau} \ln \frac{N_\tau}{N_0}$ is the population growth rate. This definition relates the relative difference of the retrospective probability from the chronological probability to fitness. $h(x)$ becomes greater than $\tau\Lambda$ if the lineage trait state $X = x$ is overrepresented in the retrospective probability relative to chronological probability and vice versa. Furthermore, if none of the states of lineage trait $X$ are overrepresented nor underrepresented, $h(x)$ becomes constant across the states and equals $\tau\Lambda$ for all $x$. The fitness landscape $h(x)$ thus represents fitness differences mapped on the lineage trait space of $X$ (see *Figure 2* and *Box 1*).

One can also define 'selection strength' using $Q_{\mathrm{rs}}(x)$ and $Q_{\mathrm{cl}}(x)$ as

$$S_{\mathrm{JF}}[X] := J[Q_{\mathrm{cl}}(X), Q_{\mathrm{rs}}(X)] = \langle h(X) \rangle_{\mathrm{rs}} - \langle h(X) \rangle_{\mathrm{cl}}, \tag{2}$$

where $J[Q_{\mathrm{cl}}(X), Q_{\mathrm{rs}}(X)] := \sum_x \left( Q_{\mathrm{cl}}(x) - Q_{\mathrm{rs}}(x) \right) \ln \frac{Q_{\mathrm{cl}}(x)}{Q_{\mathrm{rs}}(x)}$ is Jeffreys divergence, and $\langle h(X) \rangle_{\mathrm{rs}} := \sum_x h(x) Q_{\mathrm{rs}}(x)$ and $\langle h(X) \rangle_{\mathrm{cl}} := \sum_x h(x) Q_{\mathrm{cl}}(x)$ are the retrospective and chronological mean fitness for lineage trait $X$. Jeffreys divergence measures dissimilarity between two probability distributions. Therefore, $S_{\mathrm{JF}}[X]$ measures dissimilarity between the chronological and retrospective distributions caused by selection. Notably, one can link this dissimilarity to the difference in the mean fitness, as shown in *Equation 2*. Since Jeffreys divergence is non-negative, the retrospective mean fitness (mean fitness after selection) is equal to or greater than the chronological mean fitness (mean fitness before selection).

This measure of selection strength quantifies how strongly differences in the states of lineage trait $X$ correlate with the differences in lineage fitness. Therefore, one can unravel which traits correlate with lineage fitness strongly by evaluating this for traits of interest.

Likewise, we can define two alternative selection strength measures:

$$S_{\mathrm{KL}}^{(1)}[X] := D_{\mathrm{KL}}[Q_{\mathrm{cl}}(X) \| Q_{\mathrm{rs}}(X)] = \tau\Lambda - \langle h(X) \rangle_{\mathrm{cl}}, \tag{3}$$

$$S_{\mathrm{KL}}^{(2)}[X] := D_{\mathrm{KL}}[Q_{\mathrm{rs}}(X) \| Q_{\mathrm{cl}}(X)] = \langle h(X) \rangle_{\mathrm{rs}} - \tau\Lambda, \tag{4}$$

where $D_{\mathrm{KL}}[Q_{\mathrm{cl}}(X) \| Q_{\mathrm{rs}}(X)] := \sum_x Q_{\mathrm{cl}}(X) \ln \frac{Q_{\mathrm{cl}}(X)}{Q_{\mathrm{rs}}(X)}$ and $D_{\mathrm{KL}}[Q_{\mathrm{rs}}(X) \| Q_{\mathrm{cl}}(X)] := \sum_x Q_{\mathrm{rs}}(X) \ln \frac{Q_{\mathrm{rs}}(X)}{Q_{\mathrm{cl}}(X)}$ are the Kullback-Leibler divergence of the two distributions. Note that $S_{\mathrm{KL}}^{(1)}[X] + S_{\mathrm{KL}}^{(2)}[X] = S_{\mathrm{JF}}[X]$.

These three types of selection strength measures share identical properties in common: they are always non-negative and report the overall correlations between trait states and fitness. We exclusively used $S_{\mathrm{JF}}[X]$ as the selection strength measure in our previous study (*Nozoe et al., 2017*). However, $S_{\mathrm{KL}}^{(1)}[X]$, $S_{\mathrm{KL}}^{(2)}[X]$, and their difference $S_{\mathrm{KL}}^{(2)}[X] - S_{\mathrm{KL}}^{(1)}[X]$ possess their own unique biological meanings, as we detail in Results. We indeed evaluate both $S_{\mathrm{KL}}^{(1)}[X]$ and $S_{\mathrm{KL}}^{(2)}[X]$ for the empirical lineage data of various organisms and use them to unravel distinct effects of selection on fitness variances. Such meanings and roles of the different selection strength measures are clarified in this study.

Importantly, division count $D$ is also a lineage trait, and its selection strength sets the maximum bound for the selection strength of any lineage trait irrespectively of a choice of the selection strength measures as discussed in Appendix 3. Therefore, the selection strength relative to that of $D$ is bounded between 0 and 1 and evaluates how strongly the heterogeneity of $X$ correlates with the division count heterogeneity in a given cellular population. This relative measure is useful when comparing relative strength of correlations between lineage traits and fitness across conditions. In this study, we define relative selection strength as

$$S_{\mathrm{rel}}[X] := \frac{S_{\mathrm{KL}}^{(1)}[X]}{S_{\mathrm{KL}}^{(1)}[D]}, \tag{5}$$

and use it in the analysis.

All of the quantities introduced above are measurable without relying on any growth models. Thus, this cell lineage statistics framework is applicable to a wide range of single-cell lineage data.

## Box 1. A glossary of the terms

Here, we provide intuitive and illustrative explanations of the essential quantities in the cell lineage statistics and discuss their similarities and differences compared to the common usage in evolutionary biology.

**Fitness:** In evolutionary biology, *fitness* refers to the expected per capita contribution of individuals of a particular trait (usually a genotype) to the future population (*Futuyma, 2010*). For example, if a set of $N_0$ individuals with trait $X$ produce $N_1$ descendants on average in the future population, the fitness of this trait would be $N_1/N_0$. Since proliferation usually proceeds multiplicatively, the logarithm of fitness, $\ln(N_1/N_0)$, is also often referred to as 'fitness'. Analogously, in our framework we define fitness for cell lineage traits as the expected contribution of lineages with a given trait value in the future population. For each cell lineage $\sigma$, the number of cell divisions occurring along the lineage, $D(\sigma)$, is used to estimate the expected contribution of each lineage to the future population.

**Fitness landscape:** In evolutionary biology, fitness landscapes are visual representations of relationships between reproductive abilities (fitness) and genotypes (*Futuyma, 2010*), where the height along the landscape corresponds to fitness. Since "genotype space" is vast and usually difficult to construct or visualize, fitness landscapes are often referred to as a metaphorical or conceptual tool for understanding complex evolutionary processes. For practical applications, however, fitness landscapes are often mapped on a low dimensional

allele frequency space or a phenotypic space. Analogously, in our framework fitness landscapes are mapped on cell lineage trait spaces. However, they are different in that there is no assumption of genotypic differences underlying different trait states. Furthermore, the landscapes are directly measurable using cellular lineage trees and trait dynamics in each lineage.

For a cell lineage trait $X$, we define its *fitness landscape* to be a function $h(x)$ that reports the expected reproductive success of lineages having trait value $X = x$. Each lineage $\sigma$ having trait value $x$ contributes $2^{D(\sigma)}$ lineages to the future population, and by summing over lineages sharing the same trait value, we estimate the expected reproductive success of the trait and measure its fitness landscape. If differences in $X$ correlate with division count heterogeneity among cell lineages, $h(x)$ varies across the trait space of $X$; if differences in $X$ are uncorrelated with division count heterogeneity, $h(x)$ is constant over the entire space of $X$ (*Figure 2*).

**Selection:** The term *selection* refers to processes in which the frequencies of individuals with different traits change due to differences in their fitness (*Futuyma, 2010*). In evolutionary biology, selection is usually assessed based on changes in the distribution of traits between two points in times, which requires an accurate measure of fitness and a model to determine whether the observed changes were the result of trait fitness differences. In our cell lineage statistics framework, we measure selection by determining whether the observed distribution of lineage traits (i.e. the retrospective distribution) differs from the distribution expected in the absence of fitness differences (i.e. the chronological distribution). The key advantage that lineage-based analysis provides is the ability to construct explicitly the chronological distribution, which is the natural 'null hypothesis' against which selection can be tested in a model-independent manner.

**Selection strength:** $S[X]$ (i.e., $S_{\mathrm{JF}}[X]$, $S_{\mathrm{KL}}^{(1)}[X]$, or $S_{\mathrm{KL}}^{(2)}[X]$) is a quantitative measure that reports how strongly differences in the states of cell lineage trait $X$ are correlated with cell lineage fitness, taking the distributions of $X$ into account. The selection strength in our framework is measured by differences in the fitness measures or by differences between chronological and retrospective distributions (*Equations 2–4*). One can prove that these different definitions are mathematically equivalent.

The three situations depicted in *Figure 2* would help us to gain an intuitive understanding of the properties and meanings of selection strength. When $X$ is correlated with fitness, a fitness landscape $h(x)$ becomes non-uniform, as mentioned above. When the states of lineage trait $X$ are heterogeneous and distributed widely within a population, the selection causes a significant difference between chronological and retrospective distributions due to the biased representation of trait states by selection. Therefore, the selection strength becomes large ($S[X] > 0$, *Figure 2A*). In the second situation, $h(x)$ is again non-uniform, but the distribution of $x$ is narrow. In this case, there is almost no effective trait heterogeneity in the population on which selection can act. Consequently, the overall extent of selection becomes small, i.e., selection strength becomes small ($S[X] \approx 0$, *Figure 2B*). Finally, when $h(x)$ is uniform over the observed state of $x$, selection can neither overrepresent nor underrepresent any states, no matter how the trait $x$ distributes in a population. Therefore, the chronological and retrospective distributions become identical, and the selection strength becomes zero ($S[X] = 0$, *Figure 2C*).

These examples show that $S[X]$ can gauge to what extent selection acts on a lineage trait $X$, considering both shapes of fitness landscapes and distributions of lineage traits in a population. Therefore, if $X$ is a trait of interest, quantifying $S[X]$ or the relative strength of selection $S[X]/S[D]$ determines how strongly the heterogeneity of $X$ is correlated with fitness differences of cell lineages.

In evolutionary biology, various measures are used to quantify how strongly selection acts in a population of interest. For example, the 'coefficient of selection' measures a relative difference in fitness of each genotype from that of the fittest genotype (*Futuyma, 2010*). This measure is useful when considering the selection against a particular reference genotype. The overall intensity of selection in a population can be quantified by changes in mean fitness before

and after selection, variances of fitness before selection, changes in the mean of log fitness, and Jeffreys divergence between trait distributions before and after selection (**Frank, 2012**). Therefore, our definitions of selection strength follow the standard measures for the overall selection in evolutionary biology both conceptually and mathematically but are different in that the mean fitness and distributions of chronological and retrospective statistics are used.

**Cumulants:** In Results, we consider the contributions of the *cumulants* of a fitness landscape to population growth. The cumulants of a probability distribution are a set of quantities that characterize the distribution. For a discrete probability distribution $P(x)$, its cumulant generating function is defined as

$$K(\xi) := \ln E[e^{\xi X}] = \ln \sum_x e^{\xi x} P(x),$$

(6)

and the $n$-th order cumulant $\kappa_n$ is obtained by evaluating the $n$-th order derivative of $K(\xi)$ at $\xi = 0$, i.e.,

$$\kappa_n := \left. \frac{d^n K(\xi)}{d\xi^n} \right|_{\xi=0}.$$

(7)

Notably, the first few cumulants correspond to important statistical quantities. The first-order cumulant $\kappa_1$ corresponds to the mean $\langle X \rangle := E[X] = \sum_x x P(x)$, and the second-order cumulant $\kappa_2$ corresponds to the variance $\mathrm{Var}[X] := E[X^2] - E[X]^2 = \sum_x x^2 P(x) - \left(\sum_x x P(x)\right)^2$. The skewness of a distribution is usually defined as $E\left[\left(\frac{X - E[X]}{\sqrt{\mathrm{Var}[X]}}\right)^3\right]$, and this quantity can be expressed as $\kappa_3/\kappa_2^{\frac{3}{2}}$ using the third-order cumulant. Since $\kappa_2$ is positive, the sign of $\kappa_3$ determines the direction of the skewness: When $\kappa_3 > 0$, the distribution is skewed to the right with a long right tail; when $\kappa_3 < 0$, the distribution is skewed to the left with a long left tail.

## Results

### Growth rate gain and cumulant expansion of population growth rate

To quantify contributions of growth heterogeneity to population growth, we first rewrite the definition of the selection strength $S_{\mathrm{KL}}^{(1)}[X]$ (**Equation 3**) as follows:

$$\tau\Lambda = \langle h(X) \rangle_{\mathrm{cl}} + S_{\mathrm{KL}}^{(1)}[X].$$

(8)

This shows that population growth rates can be decomposed into chronological mean fitness and selection strength. In particular, when we take division count $D$ as a lineage trait, its fitness landscape is $\tilde{h}(d) = d \ln 2$ (Appendix 3), and $\langle \tilde{h}(D) \rangle_{\mathrm{cl}}/\tau$ represents the mean division rate of cellular lineages without selection. $S_{\mathrm{KL}}^{(1)}[D]/\tau$, thus, represents growth rate gain caused by the growth heterogeneity among the cellular lineages in a cellular population. Therefore, evaluating $S_{\mathrm{KL}}^{(1)}[D]/\tau\Lambda$ from single-cell lineage data provides information on the contribution of growth heterogeneity to population growth.

To further examine the connections between the disparate selection measures and elucidate their meaning, we define a function of a variable $\xi$ as

$$K_X(\xi) := \ln \langle e^{\xi h(X)} \rangle_{\mathrm{cl}} = \ln \sum_x e^{\xi h(x)} Q_{\mathrm{cl}}(x).$$

(9)

This is the cumulant generating function (cgf) of $h(x)$ with respect to the chronological distribution $Q_{\mathrm{cl}}$. We have $K_X(0) = 0$, and from the definition of fitness landscape $h(x)$ (**Equation 1**), we find

$$K_X(1) = \tau\Lambda .$$

(10)

When the radius of convergence of the Taylor expansion of $K_X(\xi)$ around $\xi = 0$ is at least 1, $K_X(1)$ can be expressed as the series using the cumulants of a fitness landscape as

$$K_X(1) = \sum_{n=1}^{\infty} \frac{\kappa_n^{(X)}}{n!},$$

(11)

**Table 1.** Relationships between $K_X(\xi)$ and quantities in cellular lineage statistics.

| | Quantities in lineage statistics | Symbol | Correspondence to $K_X(\xi)$ |
|---|---|---|---|
| Fitness | Population growth | $\tau\Lambda$ | $K_X(1)$ |
| | Chronological mean fitness | $\langle h(X)\rangle_{\mathrm{cl}}$ | $K_X'(0)$ |
| | Retrospective mean fitness | $\langle h(X)\rangle_{\mathrm{rs}}$ | $K_X'(1)$ |
| | Chronological fitness variance | $\mathrm{Var}[h(X)]_{\mathrm{cl}}$ | $K_X''(0)$ |
| | Retrospective fitness variance | $\mathrm{Var}[h(X)]_{\mathrm{rs}}$ | $K_X''(1)$ |
| Selection strength | Jeffreys divergence bet. $Q_{\mathrm{cl}}(X)$ and $Q_{\mathrm{rs}}(X)$ | $S_{\mathrm{JF}}[X]$ | $K_X'(1) - K_X'(0)$ |
| | KL divergence of $Q_{\mathrm{cl}}(X)$ from $Q_{\mathrm{rs}}(X)$ | $S_{\mathrm{KL}}^{(1)}[X]$ | $K_X(1) - K_X'(0)$ |
| | KL divergence of $Q_{\mathrm{rs}}(X)$ from $Q_{\mathrm{cl}}(X)$ | $S_{\mathrm{KL}}^{(2)}[X]$ | $K_X'(1) - K_X(1)$ |
| Growth rate gain/loss | Growth rate gain | $S_{\mathrm{KL}}^{(1)}[D]/\tau\Lambda$ | $1 - K_D'(0)/K_D(1)$ |
| | Additional growth rate loss upon perturbation | $-S_{\mathrm{KL}}^{(2)}[D]/\tau\Lambda$ | $1 - K_D'(1)/K_D(1)$ |

where $\kappa_n^{(X)} := \left.\frac{d^n K_X(\xi)}{d\xi^n}\right|_{\xi=0}$ is the $n$-th order cumulant, satisfying $\kappa_1^{(X)} = \langle h(X)\rangle_{\mathrm{cl}}$, and $\kappa_2^{(X)} = \mathrm{Var}[h(X)]_{\mathrm{cl}} = \langle h(X)^2\rangle_{\mathrm{cl}} - \langle h(X)\rangle_{\mathrm{cl}}^2$. Hence,

$$\tau\Lambda = \sum_{n=1}^{\infty} \frac{\kappa_n^{(X)}}{n!}, \tag{12}$$

which shows that population growth rates can be expanded by the cumulants of a fitness landscape for any lineage trait $X$. Additionally, since $\kappa_1^{(X)} = \langle h(X)\rangle_{\mathrm{cl}}$, comparing (*Equation 8*) and (*Equation 12*) yields

$$S_{\mathrm{KL}}^{(1)}[X] = \sum_{n=2}^{\infty} \frac{\kappa_n^{(X)}}{n!}. \tag{13}$$

Therefore, $S_{\mathrm{KL}}^{(1)}[X]$ represents the total contribution of second and higher order cumulants to population growth.

The cumulant expansion allows us to quantify the relative contributions of various statistical features of fitness distributions to population growth, such as mean, variance, and skewness. We define the cumulative contribution up to the $n$-th order cumulant as

$$W_n^{(X)} := \frac{1}{\tau\Lambda} \sum_{k=1}^{n} \frac{\kappa_k^{(X)}}{k!}, \tag{14}$$

and note that $W_n^{(X)}$ converges to 1 as $n \to \infty$. In particular, $W_1^{(X)} = \frac{\langle h(x)\rangle_{\mathrm{cl}}}{\tau\Lambda}$ and $W_2^{(X)} = \frac{1}{\tau\Lambda}\left(\langle h(X)\rangle_{\mathrm{cl}} + \frac{1}{2}\mathrm{Var}[h(X)]_{\mathrm{cl}}\right)$. We will indeed measure $W_n^{(D)}$ for various cellular species under steady and non-steady environments in the experimental sections below.

The function $K_X(\xi)$ defined in (*Equation 9*) is useful as it provides various forms of fitness and selection measures by simple algebraic calculation, as shown in *Table 1*. In general, evaluating $K_X(\xi)$ and its derivatives at $\xi = 0$ and $\xi = 1$ gives the information of chronological and retrospective statistics, respectively (Appendix 3). Therefore, $K_X(\xi)$ contains complete information on the fitness distributions in both chronological and retrospective statistics.

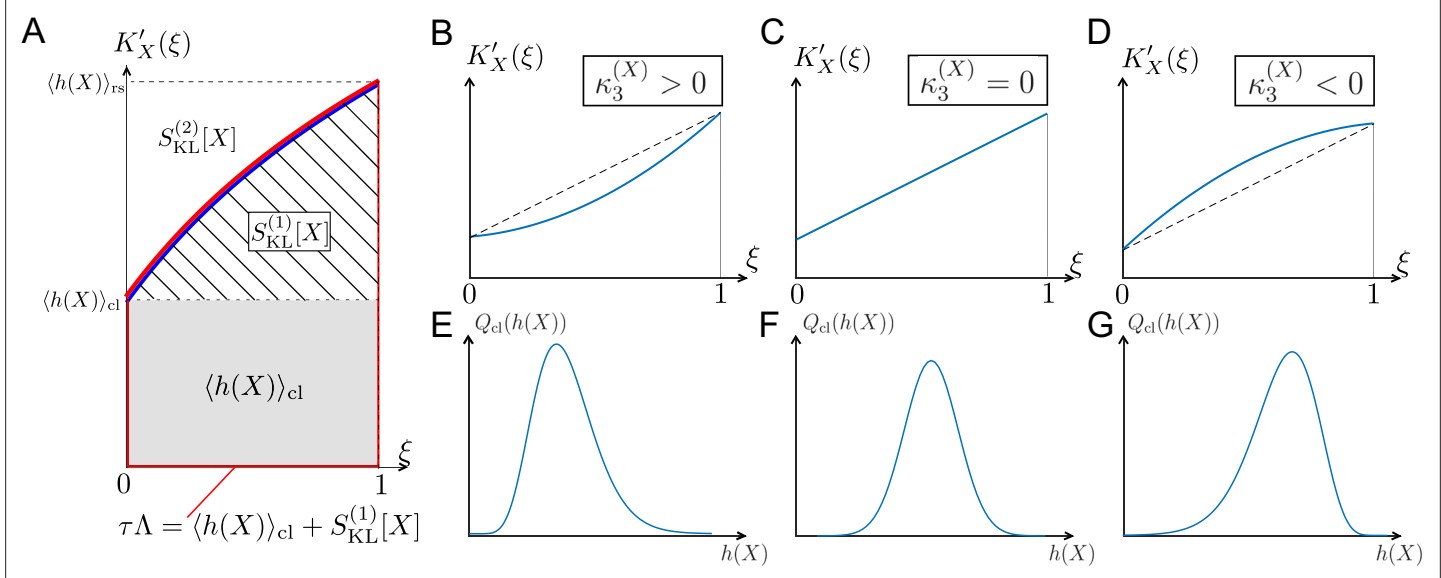

**Figure 3.** Relationships among chronological distributions' shape and selection strength measures. (**A**) Graphical representation of various fitness and selection strength measures by $K'_X(\xi)$-plot. Blue curve represents $K'_X(\xi)$. The area between the horizontal axis and $K'_X(\xi)$ in the interval $0 \le \xi \le 1$ outlined in red corresponds to population growth $\tau\Lambda$. The gray and hatched regions correspond to $\langle h(X) \rangle_{\mathrm{cl}}$ and $S^{(1)}_{\mathrm{KL}}[X]$, respectively. The area between $K'_X(\xi)$ and $y = \langle h(X) \rangle_{\mathrm{rs}}$ corresponds to $S^{(2)}_{\mathrm{KL}}[X]$. (**B-D**) Representative shapes of $K'_X(\xi)$ depending on $\kappa^{(X)}_3$. Assuming that the contributions from fourth or higher-order cumulants are negligible, $K'_X(\xi)$ becomes convex downward when $\kappa^{(X)}_3 > 0$ (**B**); a straight line when $\kappa^{(X)}_3 = 0$ (**C**); and convex upward when $\kappa^{(X)}_3 < 0$ (**D**). (**E-G**) Relationships between third-order fitness cumulant and skewness of chronological distribution $Q_{\mathrm{cl}}(h)$.

## Difference in the selection strength measures reveals the effect of selection on fitness variance

The difference between the two selection strength measures $S^{(1)}_{\mathrm{KL}}[X]$ and $S^{(2)}_{\mathrm{KL}}[X]$ is determined by the higher order cumulants by the relation

$$S^{(2)}_{\mathrm{KL}}[X] - S^{(1)}_{\mathrm{KL}}[X] = \sum_{n=3}^{\infty} \frac{\kappa^{(X)}_n}{n!}(n-2) \tag{15}$$

(Appendix 3). When fourth or higher order cumulants are negligible, the third-order fitness cumulant $\kappa^{(X)}_3$, that is the skewness of fitness distribution, determines which selection strength measure is greater.

The relations among the fitness and selection strength measures can be graphically depicted by plotting $K'_X(\xi)$ in the interval $0 \le \xi \le 1$ (**Figure 3A**). $S^{(1)}_{\mathrm{KL}}[X]$ corresponds to the area between $y = \langle h(X) \rangle_{\mathrm{cl}}$ and $y = K'_X(\xi)$; and $S^{(2)}_{\mathrm{KL}}[X]$ corresponds to the area between $y = K'_X(\xi)$ and $y = \langle h(X) \rangle_{\mathrm{rs}}$ (**Figure 3A**). Therefore, the skewness of fitness distribution primarily determines the convexity of $K'_X(\xi)$ (**Figure 3B–G**).

The difference between the two selection strength measures can reveal the effect of selection on fitness variances. The slope of the tangent lines to $K'_X(\xi)$ at $\xi = 0$ and $1$ corresponds to the chronological and retrospective fitness variances, respectively (**Table 1**). Therefore, when $K'_X(\xi)$ is convex upward in the interval $0 \le \xi \le 1$ ($\kappa^{(X)}_3 < 0$, i.e., $S^{(1)}_{\mathrm{KL}}[X] > S^{(2)}_{\mathrm{KL}}[X]$, as in **Figure 3D**), the effect of selection is to decrease the lineage fitness variance in the retrospective distribution relative to the chronological distribution, whereas if $K'_X(\xi)$ is convex downward ($\kappa^{(X)}_3 > 0$, i.e., $S^{(1)}_{\mathrm{KL}}[X] < S^{(2)}_{\mathrm{KL}}[X]$, as in **Figure 3B**), selection increases the fitness variance. We indeed find cases of both kinds of behavior in the experimental lineage data, as will be seen below. Therefore, one can probe the effect of selection on fitness variances by comparing the two selection strength measures $S^{(1)}_{\mathrm{KL}}[X]$ and $S^{(2)}_{\mathrm{KL}}[X]$.

Significant differences between $S^{(1)}_{\mathrm{KL}}[X]$ and $S^{(2)}_{\mathrm{KL}}[X]$ indicate non-negligible contributions of higher-order cumulants. In such circumstances, the fitness distributions are far from Gaussian with significant skews or multiple peaks. Therefore, higher-order cumulants can also be used to probe the existence of sub-populations in cellular populations.

## Population growth rate under fitness perturbations

We mentioned above that the selection strength measure $S_{\text{KL}}^{(1)}[D]$ represents growth rate gain caused by fitness heterogeneity. Likewise, another selection strength measure $S_{\text{KL}}^{(2)}[D]$ represents a different consequence of fitness heterogeneity, that is, additional loss of growth rate under fitness perturbations.

From (**Equation 1**), and taking division count as a lineage trait, one can express population growth rate as

$$\Lambda = \frac{1}{\tau} \ln \sum_d e^{\tilde{h}(d)} Q_{\text{cl}}(d). \tag{16}$$

We now consider the response of population growth rate to perturbations that cause lineage fitness to change from $D(\sigma)\ln 2$ to $(1-\epsilon)D(\sigma)\ln 2$, and rewrite the population growth rate as

$$\Lambda(\epsilon) := \frac{1}{\tau} \ln \sum_d e^{(1-\epsilon)\tilde{h}(d)} Q_{\text{cl}}(d). \tag{17}$$

We have $\Lambda(0) = \Lambda$, and note that $\Lambda(\epsilon) = \frac{1}{\tau} K_D(1-\epsilon)$ from (**Equation 9**). Differentiating $\Lambda(\epsilon)$ with respect to $\epsilon$, and evaluating at $\epsilon = 0$, we find

$$\left.\frac{d\Lambda(\epsilon)}{d\epsilon}\right|_{\epsilon=0} = -\frac{\langle \tilde{h}(D) \rangle_{\text{rs}}}{\tau} \tag{18}$$

(see Appendix 3). This relation shows that the change of population growth rate for small $\epsilon$ is proportional to the retrospective mean fitness of the unperturbed population. Since $\langle \tilde{h}(D) \rangle_{\text{rs}} = \tau \Lambda + S_{\text{KL}}^{(2)}[D]$ (**Equation 4**), the relative change of population growth rate is

$$\left.\frac{1}{\Lambda}\frac{d\Lambda(\epsilon)}{d\epsilon}\right|_{\epsilon=0} = -\left(1 + \frac{S_{\text{KL}}^{(2)}[D]}{\tau\Lambda}\right). \tag{19}$$

Therefore, a population with higher selection strength will exhibit a greater change in population growth rate upon perturbation. The selection strength measure $S_{\text{KL}}^{(2)}[D]$ represents additional loss of population growth rate due to division count heterogeneity before perturbation.

As we see below, one manifestation of $\epsilon$ occurs via a cell removal operation. Consider the removal of a branch in the genealogical tree just after each cell division with the probability of $1 - 2^{-\epsilon}$ ($\epsilon > 0$) (**Figure 4A**). In this case, the probability that a cell remains in the population after a cell division is $2^{-\epsilon}$, and the growth of cell lineages that originally divided $d$ times will be effectively reduced by the factor $(2^{-\epsilon})^d$. Consequently, the number of cell lineages that reach the end time point will also be effectively reduced from $N_0 \left(\sum_d 2^d Q_{\text{cl}}(d)\right)$ to $N_0 \left(\sum_d 2^{(1-\epsilon)d} Q_{\text{cl}}(d)\right)$. Therefore, the population growth rate under this branch removal operation is given by (**Equation 17**), and the relative change of population growth rate is

$$\frac{\Delta\Lambda}{\Lambda} := \frac{\Lambda(\epsilon) - \Lambda}{\Lambda} = -\left(1 + \frac{S_{\text{KL}}^{(2)}[D]}{\tau\Lambda}\right)\epsilon + O(\epsilon^2). \tag{20}$$

We validated this relation by simulating population growth with and without the cell removal operation (**Figure 4B–E** and **Figure 4—figure supplement 1**). The result confirmed that the relative changes of population growth rates by the probabilistic removal of cells followed $-\left(1 + \frac{S_{\text{KL}}^{(2)}[D]}{\tau\Lambda}\right)\epsilon$ in all the conditions (**Figure 4C–E**). We also tested this relation for cell populations with positive mother-daughter correlations of division intervals, which are often found for eukaryotic cells (**Nozoe and Kussell, 2020**; **Seita et al., 2021**; **Mosheiff et al., 2018**; **Kuchen et al., 2020**). We confirmed that the response relation was valid irrespectively of the strength of mother-daughter correlations (**Figure 4—figure supplement 1**), which shows that the relation is general and independent of the specific dynamics of the cell division process.

## Applications to models

In Appendices 1 and 2, we calculate the exact form of $K_D(\xi)$ for analytically-tractable models. We derive chronological and retrospective mean fitness, selection strength, and the cumulants of fitness landscapes from $K_D(\xi)$ to observe how the framework works. In particular, we show the analytical

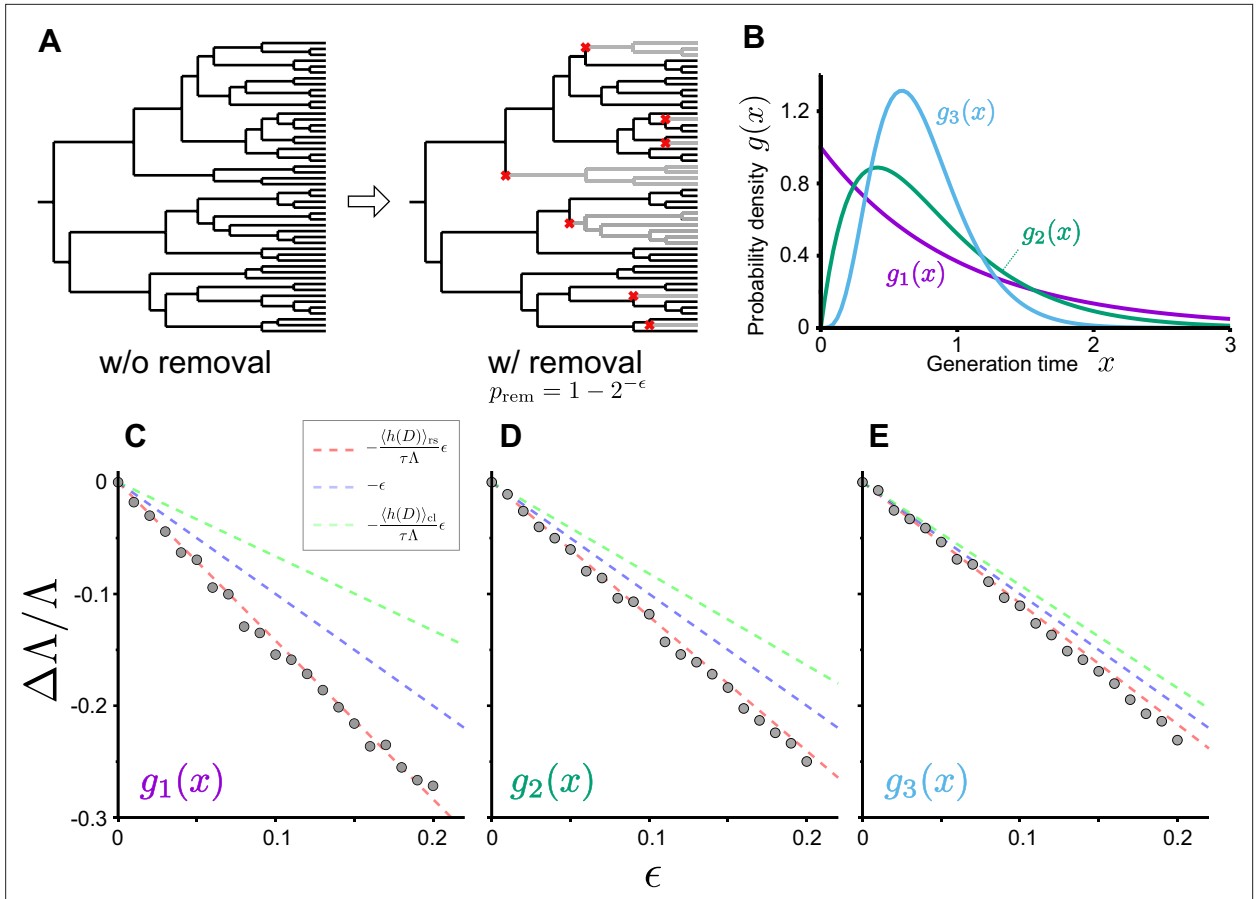

**Figure 4.** Population growth rate response to cell removal perturbation. (**A**) Scheme of random cell removal. Here, we consider the situation where cells were removed probabilistically after each cell division. Red crosses represent cell removal positions in the tree. The lineages after cell removal points disappear from the tree. Consequently, the number of cells at the end time point decreases. (**B**) Generation time distributions used in the simulation. We assumed that cellular generation time follows gamma distributions in the simulation. We set the shape parameter to either 1 ($g_1(x)$), 2 ($g_2(x)$), or 5 ($g_3(x)$). (**C-E**) Population growth rate changes by cell removal perturbation. Gray points show the relative changes in population growth rate $\Delta\Lambda/\Lambda := (\Lambda(\epsilon) - \Lambda(0))/\Lambda(0)$. Cell removal probability was set to $1 - 2^{-\epsilon}$ in each condition of perturbation strength $\epsilon$. Broken red lines represent the theoretical prediction $\Delta\Lambda/\Lambda \approx -\frac{\langle \tilde{h}(D) \rangle_{rs}}{\tau\Lambda}\epsilon = -\left(1 + \frac{S^{(2)}_{KL}[D]}{\tau\Lambda}\right)\epsilon$. The lines of $\Delta\Lambda/\Lambda = -\epsilon$ (blue) and $-\frac{\langle \tilde{h}(D) \rangle_{cl}}{\tau\Lambda}\epsilon$ (green) are shown for reference. The generation time distributions used in the simulation are $g_1(x)$ for C, $g_2(x)$ for D, and $g_3(x)$ for E.

The online version of this article includes the following figure supplement(s) for figure 4:

**Figure supplement 1.** Response of population growth rate to cell removal perturbation with positive mother-daughter correlations of generation time.

calculation for a cellular population in which cells divide with gamma-distributed uncorrelated inter-division times in Appendix 2 to understand the effect of inherent stochasticity on population growth. This analysis yields two conclusions: (1) Unlike the central limit theorem, the contribution of higher-order cumulants to population growth remains even in the long-term limit, and (2) the shape of the generation time distribution influences the cell population's long-term growth rate by constantly introducing selection within the population. Therefore, the details of inherent stochasticity of interdivision times are essential for the long-term population growth rate.

## Experimental evaluation of contributions of growth heterogeneity to population growth

Next, we apply this framework of cell lineage statistics to experimental single-cell lineage data of various organisms. The list includes bacterial cells (*Escherichia coli* and *Mycobacterium smegmatis*), unicellular eukaryotic cells (*Schizosaccharomyces pombe*), and mammalian cancer cells (L1210 mouse leukemia cells). This analysis aims to unravel whether the extent of growth rate gain from growth

**Table 2.** Summary of cellular species, culture conditions, and observation setup used in the experiments in *Figure 5*.

| Species | Label | Strain | Medium | Temperature (°C) | Device | |
|---|---|---|---|---|---|---|
| *E. coli* | rpoS-mcherry glucose_30°C | MG1655 F3 rpoS-mcherry / pUA66-PrpsL-gfp | M9 minimal medium +0.2%(w/v) glucose +1/2 MEM amino acids solution (Sigma) | 30 | Microchamber array | This study |
| *E. coli* | rpoS-mcherry glucose_37°C | MG1655 F3 rpoS-mcherry / pUA66-PrpsL-gfp | M9 minimal medium +0.2%(w/v) glucose +1/2 MEM amino acids solution (Sigma) | 37 | Microchamber array | This study |
| *E. coli* | rpoS-mcherry glycerol_37°C | MG1655 F3 rpoS-mcherry / pUA66-PrpsL-gfp | M9 minimal medium +0.1%(v/v) glycerol +1/2 MEM amino acids solution (Sigma) | 37 | Microchamber array | This study |
| *E. coli* | f3nw -sm | F3NW | M9 minimal medium +0.2%(w/v) glucose +1/2 MEM amino acids solution (Sigma)+0.1mM Isopropyl β-D-1 thiogalactopyranoside (IPTG) | 37 | Agar pad | *Nozoe et al., 2017* |
| *E. coli* | f3nw +sm | F3NW | M9 minimal medium +0.2%(w/v) glucose +1/2 MEM amino acids solution (Sigma)+0.1 mM Isopropylβ-D-1 thiogalactopyranoside (IPTG)+100 µg/ml streptomycin | 37 | Agar pad | *Nozoe et al., 2017* |
| *E. coli* | f3ptn001 -sm | F3/pTN001 | M9 minimal medium +0.2%(w/v) glucose +1/2 MEM amino acids solution (Sigma)+0.1 mM Isopropylβ-D-1 thiogalactopyranoside (IPTG) | 37 | Agar pad | *Nozoe et al., 2017* |
| *E. coli* | f3ptn001+sm | F3/pTN001 | M9 minimal medium +0.2%(w/v) glucose +1/2 MEM amino acids solution (Sigma)+0.1 mM Isopropylβ-D-1 thiogalactopyranoside (IPTG)+200 µg/ml streptomycin | 37 | Agar pad | *Nozoe et al., 2017* |
| *M. smegmatis* | mc$^2$155 7H9 | mc$^2$155 | Middlebrook 7H9 medium +0.5% albumin +0.2% glucose +0.085% NaCl+0.5% glycerol +0.05% Tween-80 | 37 | Membrane cover | *Wakamoto et al., 2013* |
| *S. pombe* | EMM28 | HN0025 | Edinburgh minimal medium +2% (w/v) glucose | 28 | Mother machine | *Nakaoka and Wakamoto, 2017* |
| *S. pombe* | EMM30 | HN0025 | Edinburgh minimal medium +2%(w/v) glucose | 30 | Mother machine | *Nakaoka and Wakamoto, 2017* |
| *S. pombe* | EMM32 | HN0025 | Edinburgh minimal medium +2%(w/v) glucose | 32 | Mother machine | *Nakaoka and Wakamoto, 2017* |
| *S. pombe* | EMM34 | HN0025 | Edinburgh minimal medium +2%(w/v) glucose | 34 | Mother machine | *Nakaoka and Wakamoto, 2017* |
| *S. pombe* | YE28 | HN0025 | Yeast extract medium +3%(w/v) glucose | 28 | Mother machine | *Nakaoka and Wakamoto, 2017* |
| *S. pombe* | YE30 | HN0025 | Yeast extract medium +3%(w/v) glucose | 30 | Mother machine | *Nakaoka and Wakamoto, 2017* |
| *S. pombe* | YE34 | HN0025 | Yeast extract medium +3%(w/v) glucose | 34 | Mother machine | *Nakaoka and Wakamoto, 2017* |
| L1210 mouse leukemia cell | L1210 RPMI-1640 | L1210 (ATCC CCL-219) | RPMI-1640 medium (Wako)+10% fetal bovine serum (Biosera) under 5% $CO_2$ atmosphere | 37 | Mother machine | *Seita et al., 2021* |

**Table 3.** Summary of the data used in the analysis in *Figure 5*.
$t_{\text{start}}$ and $t_{\text{end}}$ are the start and end times for the analysis time window $\tau$.

| Species | label | $\tau$(hr) | $t_{\text{start}}$(hr) | $t_{\text{end}}$(hr) | $N_0$ | $N_\tau$ |
|---|---|---|---|---|---|---|
| *E. coli* | rpoS-mcherry glucose_37°C | 5 | 0.95 | 5.95 | 163 | 3989 |
| *E. coli* | rpoS-mcherry glucose_30°C | 8 | 0.95 | 8.95 | 197 | 6173 |
| *E. coli* | rpoS-mcherry glycerol_37°C | 6.5 | 0.95 | 7.45 | 253 | 5825 |
| *E. coli* | f3nw-sm | 5 | 0 | 5 | 305 | 4343 |
| *E. coli* | f3nw +sm | 5 | 0 | 5 | 291 | 3164 |
| *E. coli* | f3ptn001-sm | 5 | 0 | 5 | 984 | 9229 |
| *E. coli* | f3ptn001+sm | 5 | 0 | 5 | 977 | 7429 |
| *M. smegmatis* | mc$^2$155 7H9 | 10 | 1.75 | 11.75 | 39 | 311 |
| *S. pombe* | EMM28 | 167 | 0 | 167 | 1148 | - |
| *S. pombe* | EMM30 | 131 | 0 | 131 | 963 | - |
| *S. pombe* | EMM32 | 123.5 | 0 | 123.5 | 883 | - |
| *S. pombe* | EMM34 | 152 | 0 | 152 | 1078 | - |
| *S. pombe* | YE28 | 108 | 0 | 108 | 1177 | - |
| *S. pombe* | YE30 | 90 | 0 | 90 | 866 | - |
| *S. pombe* | YE34 | 78 | 0 | 78 | 863 | - |
| L1210 mouse leukemia cell | L1210 RPMI-1640 | 60 | 0 | 60 | 474 | - |

heterogeneity depends on the organisms and environments under constant growth conditions. As summarized in *Tables 2 and 3*, we used cellular lineage data newly obtained in this study as well as other existing datasets (*Nozoe et al., 2017*; *Wakamoto et al., 2013*; *Nakaoka and Wakamoto, 2017*; *Seita et al., 2021*). The *E. coli* and *S. pombe* data include several culture conditions to compare cumulants' contributions to population growth across environments. The *E. coli* data were obtained using either agarose pad or the microchamber array microfluidic device, yielding genealogical tree information such as the one shown in *Figure 1*. The *S. pombe* and L1210 cell data were obtained with mother machine microfluidic devices (*Wang et al., 2010*), which provide isolated cell lineage information but discard tree information due to its cell exclusion scheme. We assumed that these isolated cell lineages would follow chronological statistics and evaluated chronological distributions and selection strength according to the method described in Materials and methods. All the data analyzed in this section were taken from cell populations growing at approximately constant rates.

First, we evaluated the first-order cumulants' contributions $W_1^{(D)} = \frac{\kappa_1^{(D)}}{\tau\Lambda} = \frac{\langle\tilde{h}(D)\rangle_{\text{cl}}}{\tau\Lambda}$ (*Equation 14*), finding that $W_1^{(D)} < 1$ for all the samples and conditions (*Figure 5A*). This result confirms that the chronological mean fitness cannot fully account for the population growth rates. This means that the division count heterogeneity present even in constant environments contributes to increasing the population growth rate. However, the extent of the contributions was different: $W_1^{(D)}$ for *S. pombe* was consistently closer to 1 than those for the other cell types except one condition (EMM, 34 °C), suggesting that *S. pombe*'s growth is less heterogeneous under most culture conditions.

We next evaluated $W_2^{(D)} = \frac{\kappa_1^{(D)}+\kappa_2^{(D)}/2}{\tau\Lambda}$, finding that $W_2^{(D)} \approx 1$ for most of the conditions (*Figure 5A*). This result indicates small contributions of the third or higher-order cumulants to population growth. Consistent with this result, $S_{\text{KL}}^{(1)}[D]$ and $S_{\text{KL}}^{(2)}[D]$ were almost identical in most conditions (*Figure 5B*). Note that $S_{\text{KL}}^{(2)}[D] - S_{\text{KL}}^{(1)}[D]$ depends only on the third or higher order cumulants (*Equation 15*). The chronological distributions $Q_{\text{cl}}(D)$ of these samples were nearly symmetric in most cases; however, under the conditions where the deviations of $W_2^{(D)}$ from 1 are larger, such as *S. pombe* in EMM medium and L1210, the distributions were skewed slightly (*Figure 5C–E* and *Figure 5—figure supplement 1*). Such distribution skew was reflected in the convexity directions of $K_D'(\xi)$-plots (*Figure 5F–H* and *Figure 5—figure supplement 2*). These results imply that cellular populations of *S. pombe* in EMM

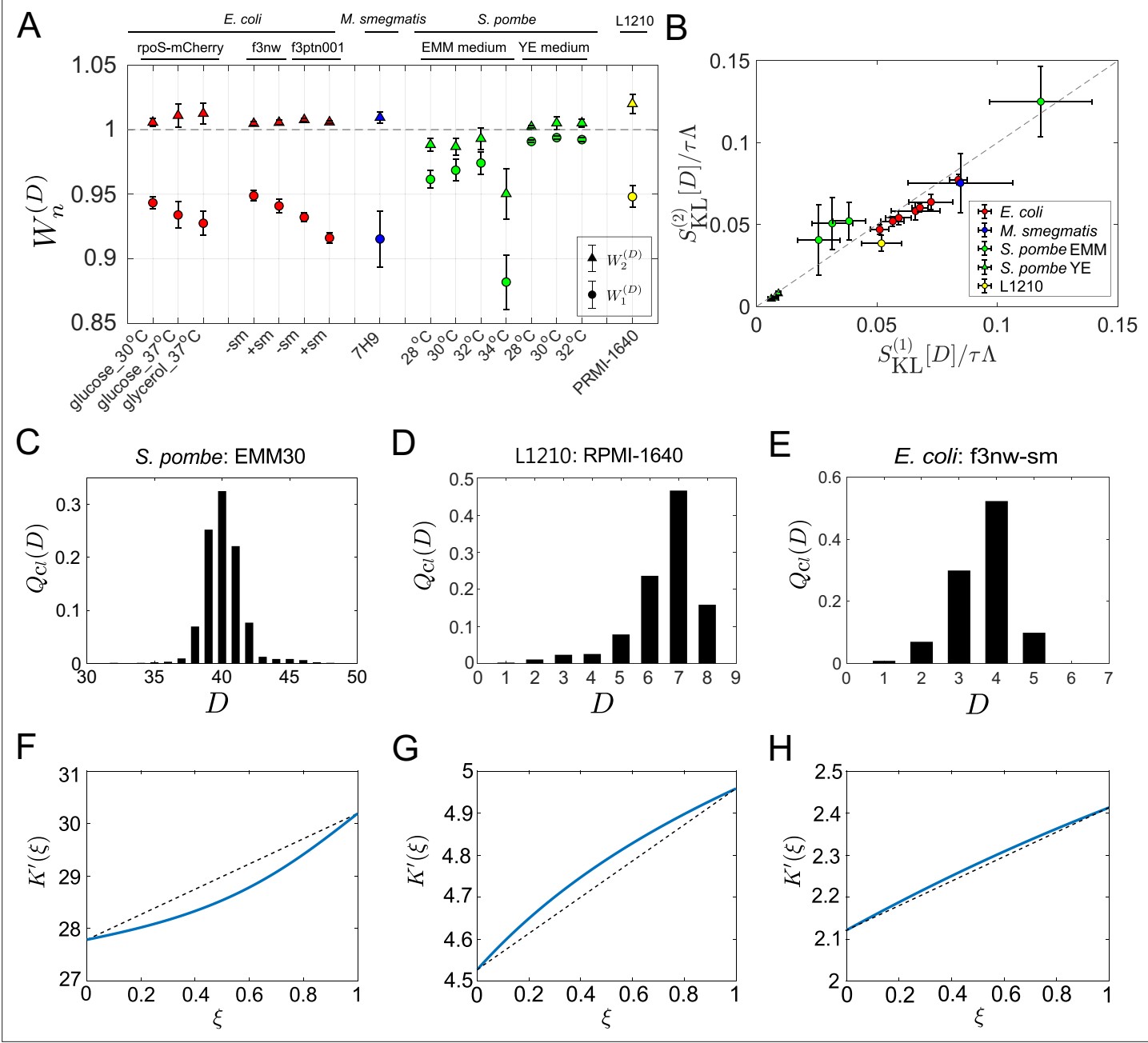

**Figure 5.** Application of cell lineage statistics to experimental data. (**A**) Contributions of the cumulants of a fitness landscape to population growth. $W_1^{(D)}$ and $W_2^{(D)}$ were evaluated for the experimental cell lineage data from *E. coli* (red), *M. smegmatis* (blue), *S. pombe* (green), and L1210 mouse leukemia cells (yellow). The *E. coli* rpoS-mcherry data were newly obtained in this study (see Materials and methods). The other data were taken from literature: *E. coli* f3nw and f3ptn001 from ***Nozoe et al., 2017***; *M. smegmatis* from ***Wakamoto et al., 2013***; *S. pombe* from ***Nakaoka and Wakamoto, 2017***; and L1210 from ***Seita et al., 2021***. Circles and triangles represent $W_1^{(D)}$ and $W_2^{(D)}$, respectively. Error bars represent the two standard deviation ranges estimated by resampling the cellular lineages (see Materials and methods). (**B**) Relationship between $S_{\mathrm{KL}}^{(1)}[D]/\tau\Lambda$ and $S_{\mathrm{KL}}^{(2)}[D]/\tau\Lambda$. Colors correspond to the cellular species as in A. The *S. pombe* data were further categorized into two groups: Circles for the EMM conditions; and triangles for the YE conditions. (**C-E**) Representative chronological distributions of division count, $Q_{\mathrm{cl}}(D)$. (**F-H**) Graphical representation of $K'_D(\xi)$. F for *S. pombe* EMM30; G for L1210 RMPI-1640; and H for *E. coli* f3nw-sm.

The online version of this article includes the following figure supplement(s) for figure 5:

**Figure supplement 1.** Chronological distributions of division count, $Q_{\mathrm{cl}}(D)$.

**Figure supplement 2.** Graphical representation of $K'_D(\xi)$.

medium and of L1210 contain small subpopulations that follow distinct division statistics. In fact, it was previously demonstrated that the L1210 cell populations contain slow-cycling cell lineages that can survive for longer durations under exposure to an anticancer drug (*Seita et al., 2021*). Therefore, this analysis confirms that the differences in the two strength measures can be used for detecting subpopulations in cellular populations.

In *S. pombe* EMM medium conditions, $K'_D(\xi)$ was convex downward in the interval $0 \leq \xi \leq 1$ except for EMM 34°C (*Figure 5F* and *Figure 5—figure supplement 2*). Therefore, under certain conditions selection can increase fitness variance in the retrospective distributions relative to chronological distributions among cellular lineages.

## The contributions of higher order cumulants become significant in the regrowth from a late stationary phase

We further applied the framework to the cell lineage data of *E. coli* populations regrowing from an early or late stationary phase. This analysis aims to uncover how strongly selection occurs upon environmental changes and whether the selection strength can differ under identical conditions depending on the conditions before regrowth. To conduct time-lapse observations of regrowing cell populations, we used a microfluidic device equipped with microchambers etched on a glass coverslip. We sampled *E. coli* cells either from an early or late stationary phase batch culture and enclosed the cells into the microchambers by a semipermeable membrane (*Inoue et al., 2001*; *Hashimoto et al., 2016*). We switched flowing media from stationary-phase conditioned medium to fresh medium at the start of time-lapse measurements and recorded the growth and division of individual cells (*Figure 6A*, see Materials and Methods).

The growth curves reconstructed by counting the number of cells at each time point showed lags in regrowth (*Figure 6B*). The lag time was shorter for the populations from the early stationary phase. The lineage tree structures in the cell populations were markedly different between the conditions (*Figure 6C and D*). The tree structures were more uniform for the early stationary phase sample with multiple divisions in most cell lineages (*Figure 6C*), whereas those for the late stationary phase sample were more heterogeneous, with 90% of cells showing no divisions within the observation time (*Figure 6D*).

We analyzed these data and found $W_1^{(D)} = 0.95 \pm 0.02$ for the population from the early stationary phase and $W_1^{(D)} = 0.27 \pm 0.04$ for the population from the late stationary phase (*Figure 6E*). Therefore, the chronological mean fitness, $\langle \tilde{h}(D) \rangle_{\mathrm{cl}}$, explains only 27% of the growth rate of the population regrowing from the late stationary phase. In other words, significantly strong selection occurred in the regrowth from the late stationary phase. We also found that $W_2^{(D)} \approx 1$ for the population from the early stationary phase, as observed for the *E. coli* populations growing at constant rates. In contrast, $W_2^{(D)}$ for the population from the late stationary phase was $0.61 \pm 0.04$, and $W_n^{(D)}$ converged to 1 only after taking the cumulants up to approximately 10th-order into account (*Figure 6E*). This indicates a skew of the fitness distribution and validates the existence of subpopulations following distinct division statistics in the population from the late stationary phase in this time scale of regrowth (*Figure 6F*). Reflecting the extreme skew to the right of the chronological distributions $Q_{\mathrm{cl}}(D)$ (*Figure 6F*), $S_{\mathrm{KL}}^{(1)}[D]$ was significantly greater than $S_{\mathrm{KL}}^{(2)}[D]$ for the late stationary phase sample (*Figure 6G*).

These results indicate that the levels of selection in the regrowing processes strongly depend on the durations under stationary phase conditions. Therefore, the ability to quickly resume growth under favorable conditions is gradually lost in most cells in the stationary phase; only a fraction of cells in the population can contribute to the future cell population. However, we also remark that preserving such non-growing cell lineages can be beneficial when cell populations are exposed to harsh environments in unpredictable manners (*Kussell and Leibler, 2005*).

## Lineage statistics reveal condition-dependent fitness landscapes and selection strength for a growth-regulating sigma factor

RpoS is a sigma factor that controls the transcription of a large set of genes (10% of the genome) in *E. coli* (*Battesti et al., 2011*). High RpoS expression usually correlates with growth suppression; RpoS is induced when cells enter stationary phases or encounter stress conditions, such as starvation, low pH, oxidative stress, high temperature, or osmotic stress. Elevated RpoS expression provokes the intracellular programs to shut down growth and resist the stress (*Battesti et al., 2011*). However, it

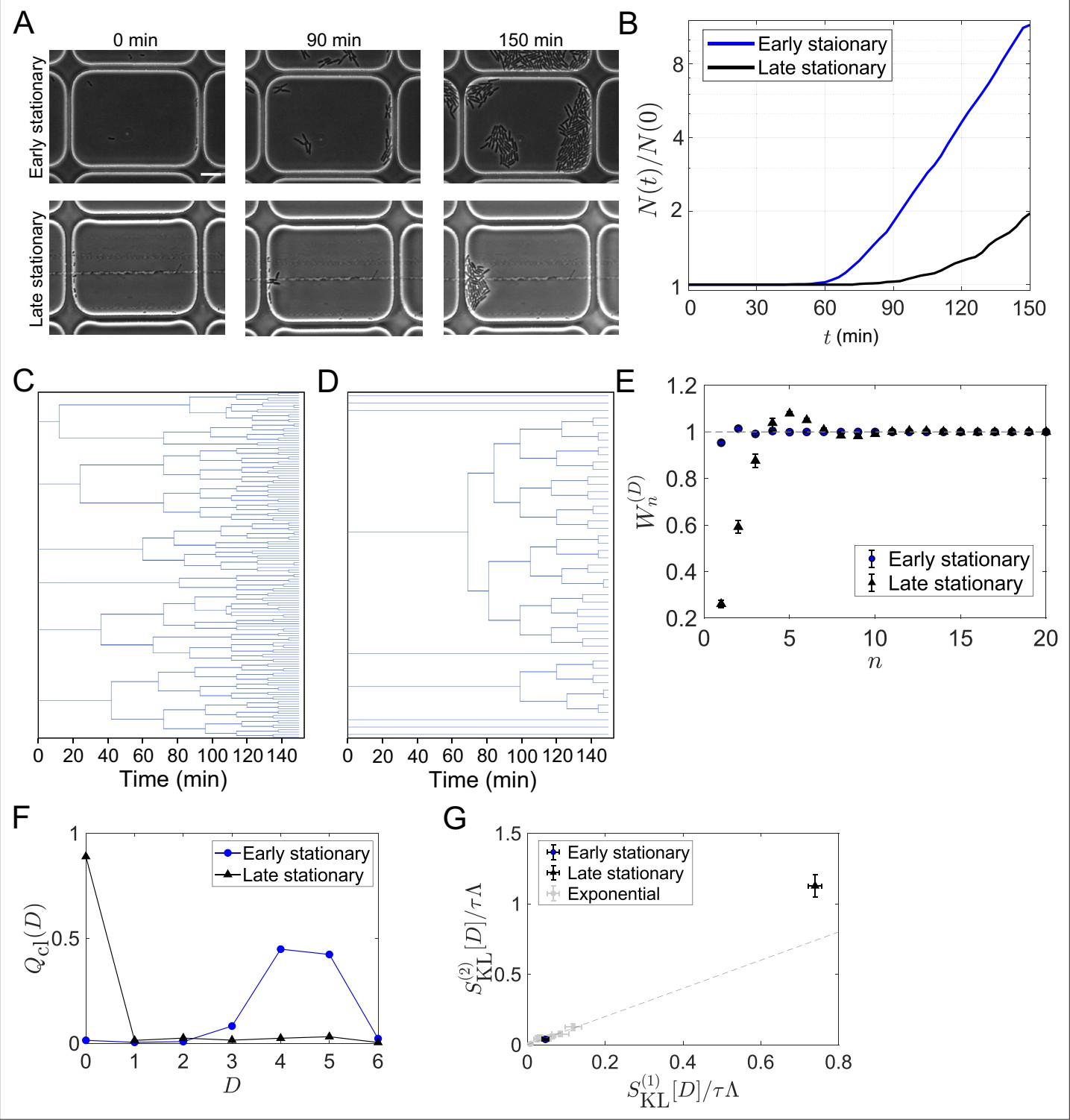

**Figure 6.** Strong selection in the *E.coli* population regrowing from a late stationary phase. (**A**) Time-lapse images. Cellular regrowing dynamics from early and late stationary phases were observed by time-lapse microscopy. Cells were enclosed in the microchambers etched on coverslips. The top three images show representative images of the cells from an early stationary phase. The bottom three images show the cells from a late stationary phase. Scale bar, 5 μm. (**B**) Population dynamics. The number of cells at each time point normalized by the initial cell number ($N(t)/N(0)$) was plotted against time $t$ was 307 for the early stationary sample and 295 for the late stationary sample. (**C, D**) Representative cellular lineage trees in the regrowing kineics from the early stationary phase (**C**) and the late stationary phase (**D**). The trees correspond to the time-lapse images in A. (**E**) Cumulative contributions of the cumulants of the fitness landscape $h(D)$ to population growth. Error bars represent the two standard deviation ranges estimated by resampling

*Figure 6 continued on next page*

*Figure 6 continued*

the cellular lineages (see Materials and methods). (**F**) Chronological distributions of division count $Q_{cl}(D)$. (**G**) Relationships between $S_{KL}^{(1)}[D]$ and $S_{KL}^{(2)}[D]$. The blue and black points show the results for the early stationary phase sample and the late stationary phase sample, respectively. Gray points represent the results for the cell populations growing at approximately constant growth rates shown in *Figure 5B*.

remains poorly understood how the continuum heterogeneity of RpoS expression levels is linked to the lineage fitness and selection in exponentially growing cellular populations. We therefore applied the lineage statistics framework to the single-cell time-lapse data of an *E. coli* strain expressing an RpoS-mCherry fusion protein from the native chromosomal locus and green fluorescent protein (GFP) from a low copy plasmid.

We quantified the time-scaled fitness landscapes $h(X)/\tau$ and relative selection strength $S_{rel}[X]$ (*Equation 5*) under three growth conditions, taking the time-averaged mean fluorescent intensity of RpoS-mCherry or GFP along each cell lineage (proxies of time-averaged intracellular concentrations) as $X$ (*Figure 7*). Since fluorescent intensity is a trait that takes continuous values, we binned the intensity values with the bin sizes around which selection strength values are relatively stable (Materials and methods). Furthermore, since the calculation of relative selection strength from empirical data always gives positive values, we compared the relative selection strength values with those calculated from the data in which the correspondences between division counts and trait values were randomized to confirm the confidence levels (*Figure 7—figure supplement 1*).

The result shows that the fitness landscapes and selection strength of RpoS expression level differ significantly among the growth conditions (*Figure 7*). Under the glucose-37°C condition, the fitness landscapes of RpoS-mCherry and GFP expression were both decreasing functions (*Figure 7A and B*). Thus, high expression of RpoS-expression and GFP in an exponentially growing population are both linked with lower lineage fitness. However, while the fitness landscape of GFP expression were nearly constant and showed significant decrease of fitness only at high expression levels, the fitness landscape of RpoS-mCherry decreased steadily in the observed expression range (*Figure 7A and B*). Consequently, the relative selection strength for RpoS-mCherry was 2.6-fold larger than that for GFP (*Figure 7C*).

Under the glucose-30°C and glycerol-37°C conditions, the fitness landscapes for RpoS-mCherry level were also decreasing functions and close to each other but significantly downshifted from that for the glucose-37°C condition (*Figure 7A*). This result reveals that cells could have different fitness for the same expression levels of RpoS, depending on the growth conditions. The selection strength for RpoS-mCherry was larger than that for GFP under the glucose-37°C and glucose-30°C conditions (*Figure 7C*), which proves that the heterogeneity of RpoS expression in a population correlates with the lineage fitness more strongly than that of GFP under those conditions. On the other hand, the relative selection strength of RpoS-mCherry under the glycerol-37°C condition was the smallest among the three conditions and not significantly different from that of GFP (*Figure 7C*). This is due to the relatively flat fitness landscapes in the central ranges of the distributions $Q_{cl}(x)$ (*Figure 7A and B*) and the smaller variations of $x$ in the population (*Figure 7D and E*). These results reveal that the continuum heterogeneity of RpoS expression level in a population does correlate with the lineage fitness, but its contribution to selection depends on growth conditions. In other words, the heterogeneity in the RpoS-mCherry expression levels can barely correlate with fitness heterogeneity under some conditions.

We also evaluated the contributions of fitness cumulants for RpoS-mCherry expression to the population growth rate. Under all the conditions, $W_1^{(X)}$ was lower than 1 (*Figure 7F–H*). Therefore, the contributions of the higher-order fitness cumulants are non-negligible. However, the deviation of $W_1^{(X)}$ from 1 for RpoS-mCherry under the glycerol-37°C condition was small (*Figure 7H*). Hence, in this growth condition, RpoS-mCherry expression barely correlated with fitness heterogeneity in the population.

Importantly, this analysis can simultaneously reveal the changes in fitness landscapes (*Figure 7A*) and chronological distributions (*Figure 7D*). Interestingly, the distributions of the RpoS-mCherry expression levels are close between the Glucose-37°C and the Glycerol-37°C conditions, but the fitness landscapes are close between the Glucose-30°C and the Glycerol-37°C conditions. These results imply that the distributions and the fitness landscapes may vary independently in different

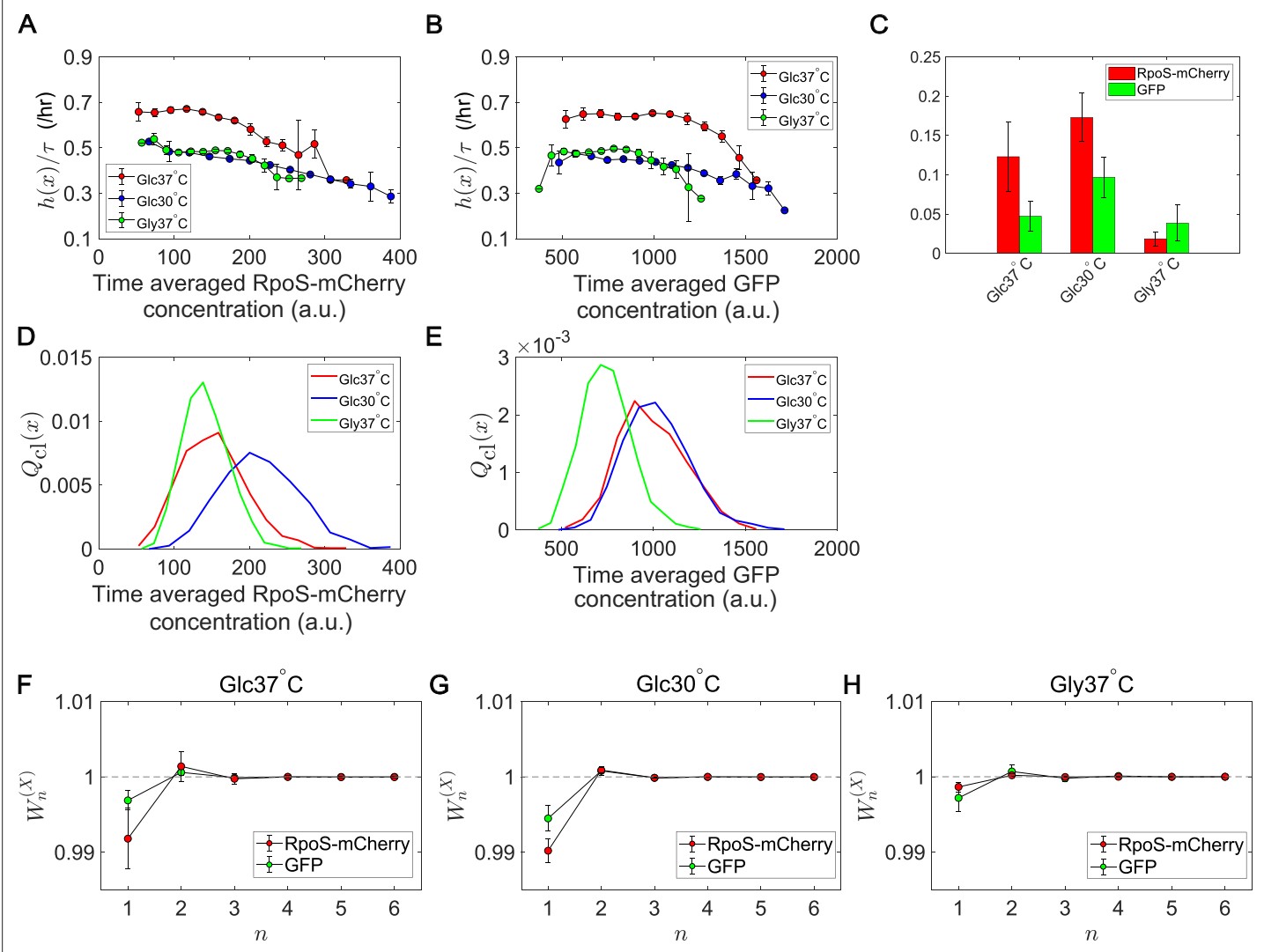

**Figure 7.** Fitness landscapes and selection strength for RpoS expression levels. (**A**) Fitness landscapes for the time-averaged concentration (mean fluorescent intensity) for RpoS-mCherry. The time-averaged mean fluorescent intensity of RpoS-mCherry was adoped as a lineage trait $X$ and changes in fitness were plotted against the trait values $x$. Fitness landscapes were scaled by the lineage length (observation duration) $\tau$. Error bars represent the two standard deviation ranges estimated by resampling the cellular lineages. (**B**) Fitness landscapes for the time-averaged concentration for GFP. The time-averaged mean fluorescent intensity of GFP was adoped as a lineage trait $X$ and changes in fitness were plotted against the trait values $x$. (**C**) Relative selection strength for the time-averaged concentrations of RpoS-mCherry (red) and GFP (green). (**D, E**) Chronological distributions $Q_{cl}(x)$ for the time-averaged concentrations of RpoS-mCherry (**D**) and GFP (**E**). (**F-H**) Cumulative contributions of fitness cumulants to population growth, $W_n^{(X)}$, assuming that $X$ is either time-averaged concentration of RpoS-mCherry (red) or time-averaged concentration of GFP (green). Error bars represent the two standard deviation ranges estimated by resampling the cellular lineages. Panel F is for the Glucose-37°C condition; Panel G for the Glucose-30°C condition; and Panel H for the Glycerol-37°C condition.

The online version of this article includes the following figure supplement(s) for figure 7:

**Figure supplement 1.** The relative selection strength values for time-averaged RpoS-mCherry and GFP fluorescence intensity compared with the randomized data.

conditions. Therefore, cells can potentially modulate the selection strength in each environment either by changing the fitness landscape or by changing the distribution of expression levels.

## Discussion

Growth and division of individual cells are intrinsically variable, which causes division count heterogeneity among cellular lineages in a population. Such heterogeneity is ubiquitous across prokaryotic

and eukaryotic cells, and its statistical properties could depend on the mechanisms and regulations determining cell division timings. Notably, division count heterogeneity influences population growth rate and, consequently, a population's survival and evolutionary success. Therefore, understanding what statistical features are produced among cellular lineages and how these features contribute to population growth is essential for unraveling each organism's survival and evolutionary strategy.

This report presents a cell lineage statistics framework to uncover the linkage between fitness distributions and population growth rate. We reveal that a population's growth rate can be expanded by the cumulants of a fitness landscape for any lineage trait. The cumulant expansion allows us to quantify the contribution of each fitness cumulant, such as variance and skewness, to population growth rate. Applying this framework to the experimental cell lineage data revealed the cumulants' contributions to population growth for various organisms and environmental conditions. In particular, higher-order cumulants became significant in the regrowth of *E. coli* from a late stationary phase. We remark that the cumulant expansion of population growth rate is valid only when all the cumulants are finite and when the Taylor expansion of $K_X(\xi)$ around $\xi = 0$ also converges at $\xi = 1$. However, all the experimental data examined in this study exhibited stable convergence, including in the regrowth condition from the late stationary phase.

An advantage of this framework is its independence from any growth and division models. The mechanisms driving the growth and division of individual cells are diverse among organisms. For example, the properties of cellular growth and division, such as whether a cell's size increases exponentially or linearly and whether cell size regulation follows sizer or adder models, could depend on cell types, organisms, and environmental conditions (*Jun et al., 2018*; *Kohram et al., 2021*). Therefore, any model assumptions restrict applicability and necessitate model validation before application. The model independence of the framework presented here comes from the definitions of two essential quantities: the chronological and retrospective probabilities. Quantifying these probabilities requires only the information of the numbers of cells at initial and end time points and of division counts on each cellular lineage. Consequently, this formalism can be applied even to non-stationary conditions without modifications. However, we also remark that this independence from the details other than cell lineage structures imposes a limitation on the framework because it cannot report any potential influences from factors such as heterogeneous environments around cells and non-quantified traits. Furthermore, the fitness landscape $h(x)$ and the relative selection strength $S_{\text{rel}}[X]$ evaluate only the correlations between the trait and fitness, not causal relationships. However, causal traits should have large selection strength values, and this framework helps narrow down the candidates for essential traits. Most importantly, division statistics is the focal information that connects molecular details underlying cellular growth and division to population growth. Regulatory mechanisms can influence population growth only by modulating the division statistics in a cellular population.

Growth heterogeneity in a cellular population plays a critical role in its adaptation and survival against stressful conditions. In antibiotic persistence, bacterial cell populations often harbor small populations of non-growing or slow-growing cells which can survive under antibiotic exposures (*Balaban et al., 2004*). Such structures of growth heterogeneity can be investigated in a unified manner by the selection strength measures introduced here. For example, the differences in $S_{\text{KL}}^{(1)}[D]/\tau\Lambda$ among organisms can reveal the distinct levels of the overall growth heterogeneity of these organisms. Furthermore, the differences between $S_{\text{KL}}^{(1)}[D]$ and $S_{\text{KL}}^{(2)}[D]$ characterize the structure of growth heterogeneity: If $S_{\text{KL}}^{(1)}[D] > S_{\text{KL}}^{(2)}[D]$, the distribution of lineage fitness is skewed negatively, and the cell population harbors small subpopulations of slow-growing cell lineages; on the contrary, if $S_{\text{KL}}^{(1)}[D] < S_{\text{KL}}^{(2)}[D]$, the population harbors small populations of fast-growing cell lineages. Untangling the linkage between the structures of growth heterogeneity and their adaptability would help us understand the adaptive strategies of various organisms.

In general, heredity is also crucial for the growth and evolution of a population. The role of the heredity of a particular trait might be unravelled by taking the correlation length as a lineage trait $X$ and quantifying its selection strength. Since the modes of heredity can also be important targets of natural selection (*Rivoire and Leibler, 2014*), such measurements might provide insights into the evolution of heredity.

We remark that the distribution of interdivision time (generation time) influences the long-term growth rate, as demonstrated by the analytical model in Appendix 2. Therefore, statistical properties of generation time, such as distribution shapes and transgenerational correlations, can contribute to

organisms' evolutionary success by constantly introducing selection within a population. Unlike the central limit theorem, the contributions of higher-order cumulants can remain even in the long-term limit. Importantly, even when cell division processes seem purely stochastic, different states in some traits might underlie these variations in generation times. In such cases, $h(x)$ and $S_{\mathrm{rel}}[X]$ for these traits can still unravel the correlations between the trait values and fitness.

This framework is applicable even to cell populations growing under non-constant environmental conditions. We indeed utilized this framework to analyze the regrowth of growth-arrested cells from the stationary phase conditions. The selection strength contributions to population growth, $S_{\mathrm{KL}}^{(1)}[D]/\tau\Lambda$, were below 10% in most cases under constant growth conditions. Nevertheless, it became over 70% in the regrowth of *E. coli* from the late stationary phase. While increased selection in non-constant environments may not be surprising itself, it is intriguing to ask how its contribution changes quantitatively depending on the conditions of environmental changes, such as nutrient upshift and downshift. The selection strength contribution in the regrowth from the early stationary phase was only 5%. This result clearly shows that how strongly selection acts in regrowing processes depends on stationary phase incubation durations. However, we also remark that the differences in the selection strength values depend on the time window and might be valid only in this time scale. Clarifying the differences in the selection strength in longer time scales requires the detail of their lag time distributions, which we did not measure in this study.

We identified the cellular populations in which selection acts to increase fitness variance in the retrospective statistics compared with the chronological statistics (*Figures 5F and 6G* and *Figure 5— figure supplement 2*). When a decrease in fitness variance by selection is mentioned in evolutionary biology, an upper bound and inheritance of fitness across the generations of individuals are usually assumed. In such circumstances, selection drives the fitness distribution toward the maximum value, and the selection eventually causes fitness variance to decrease. However, even in this process, a decrease is not assured for every step; whether selection reduces fitness variance at each step depends on the fitness distribution at that time. Likewise, whether the fitness variance increases or decreases in the retrospective distribution depends on the shape of the fitness distribution before selection, that is, chronological distribution. Such conditions are graphically recognized by the downward convexity of $K_D'(\xi)$ (*Figure 3*). When the fourth or higher order fitness cumulants are negligible, the convexity of $K_D'(\xi)$ is determined primarily by the skewness of $Q_{\mathrm{cl}}(d)$; positive skew of $Q_{\mathrm{cl}}(d)$ with a long right tail makes $K_D'(\xi)$ convex downward and $\mathrm{Var}[\tilde{h}(D)]_{\mathrm{rs}}$ greater than $\mathrm{Var}[\tilde{h}(D)]_{\mathrm{cl}}$. This consequence is intuitively understandable since the right tail of $Q_{\mathrm{cl}}(d)$ is accentuated in proportion to $e^D$ by selection, which leads to greater variance of $Q_{\mathrm{rs}}(d)$. On the other hand, when the skew is negative with the long left tail, the effect of applying $e^D$ is to diminish the tail and compress the distribution toward the fittest lineages. It is of note that greater fitness variance in the retrospective statistics is possible even in the long-term limit, as demonstrated by the model in Appendix 2.

We showed that division count heterogeneity among cellular lineages has dual facets: increasing population growth rate while sensitizing populations to perturbations. These two effects are quantitatively represented by $S_{\mathrm{KL}}^{(1)}[D]/\tau\Lambda$ and $S_{\mathrm{KL}}^{(2)}[D]/\tau\Lambda$, respectively. Therefore, the difference between these selection strength measures gauges which aspect of growth heterogeneity is more significant in the population. Even though $S_{\mathrm{KL}}^{(1)}[X]$ and $S_{\mathrm{KL}}^{(2)}[X]$ are different in general, the analysis revealed that they were nearly identical in most of the cellular populations growing at constant rates (*Figure 5*). This result might suggest that, from a practical viewpoint, the contribution of higher-order cumulants becomes negligible under steady growth conditions, and the significant difference between $S_{\mathrm{KL}}^{(1)}[X]$ and $S_{\mathrm{KL}}^{(2)}[X]$ could be used as a probe for the non-stationarity of the population growth. This speculation must be examined experimentally using various organisms and cell types across diverse environmental conditions.

This framework is premised on complete lineage tree information. However, many methods of single-cell measurements continuously exclude cells from observation areas and provide only a part of the tree information. Therefore, extending this framework so that one can infer both chronological and retrospective probabilities from incomplete tree information is an essential future research direction. In this study, we calculated the fitness landscapes and selection strength measures for the cell lineage data obtained with the mother machine devices, assuming that these cell lineages would follow the chronological statistics. Such a simple approach is not yet available for larger scale lineage tree data obtainable with the other single-cell measurement devices such as dynamics cytometer

(*Hashimoto et al., 2016*) and chemoflux (*Lambert et al., 2014*). Furthermore, it has been shown that the inference precision of population growth rate has non-monotonic dependence on the length of cell lineages obtained with mother machine devices (*Levien et al., 2020*). Even though the difficulties to overcome are present, a comprehensive framework may permit a unified treatment of cellular lineage data obtained using various single-cell measurement methods.

Phenotypic heterogeneity is widely observed in diverse cellular systems, including both prokaryotic and eukaryotic cells. It is often considered that phenotypic heterogeneity allows bet-hedging against unpredictable environments and promotes the survival of cellular population (*Kussell and Leibler, 2005*). However, quantitative evaluation of correlations between the traits of interest and fitness is usually an intricate problem. The cell lineage statistical framework described in this study offers a straightforward procedure applicable to any cellular genealogical data, which are now becoming increasingly available for various biological phenomena, including cancer metastasis (*Quinn et al., 2021*) and stem cell differentiation (*Filipczyk et al., 2015*; *Frieda et al., 2017*; *Chow et al., 2021*). Another important advantage of this framework is that it allows decomposing a population growth rate into chronological fitness and selection strength. It is thus intriguing to apply this framework to long-term evolutionary dynamics and quantify how the contributions of chronological mean fitness and selection underlie the transitions of population growth rate. Such analysis might clarify the crucial roles of phenotypic heterogeneity in facilitating evolution.

# Materials and methods

## Key resources table

| Reagent type (species) or resource | Designation | Source or reference | Identifiers | Additional information |
|---|---|---|---|---|
| Recombinant DNA reagent | pUA66-PrpsL-gfp (plasmid) | *Zaslaver et al., 2006* | | |
| Strain, strain background (*Escherichia coli*) | MG1655 F3 | Wakamoto lab | | MG1655ΔfliCΔfimAΔflu |
| Strain, strain background (*Escherichia coli*) | MG1655 F3 rpoS-mcherry /pUA66-P rplS-gfp | Wakamoto lab | | MG1655ΔfliCΔfimAΔflu rpoS-mcherry /pUA66-PrplS-gfp |

## Microfabrication of microchamber array

We constructed and used a microchamber array for conducting single-cell time-lapse observation under controlled environmental conditions. A microchamber is a well etched on a glass coverslip. We used two types of microchamber array. One is an array of microchamber, whose dimension is 70 μm (w) × 55 μm (h) × 1 μm (d). This microchamber has a 21-μm×7-μm pillar for supporting the membrane in the middle. We used this microchamber array for the exponential-phase experiment of *E. coli*. Another is an array of microchamber, whose dimension is 40 μm (w) × 30 μm (h) × 1 μm (d). We used this type of microchamber array for the stationary-phase-regrowth experiment in *Figure 6*. We fabricated these microchamber arrays following similar procedures described in *Hashimoto et al., 2016*; *Inoue et al., 2001*.

The photomasks for the microchamber array were created by laser drawing (DDB-201-TW, Neoark) on mask blanks (CBL4006Du-AZP, CLEAN SURFACE TECHNOLOGY). The photoresist on mask blanks was developed in NMD-3 (Tokyo Ohka Kogyo). The uncovered chromium (Cr)-layer was removed in MPM-E30 (DNP Fine Chemicals), and the remaining photoresist was removed by acetone. Lastly, the slide was rinsed in MilliQ water and air-dried.

The microchamber array was created in glass coverslips by chemical etching. First, we coated a 1,000-angstrom Cr-layer on a clean coverslip (NEO Micro glass, No. 1., 24 mm × 60 mm, Matsunami) by evaporative deposition and AZP1350 (AZ Electronic Materials) by spin-coating on the Cr-layer. We transferred the photomask patterns using a mask aligner (MA-20, Mikasa). After developing the photoresist in NMD-3 and the Cr-layer in MPM-E30, the coverslip was soaked in buffered hydrofluoric acid solution (110-BHF, Morita Kagaku Kogyo) for 14 minutes 20 seconds at 23°C for glass etching. The etching reaction was stopped by soaking the coverslip in milliQ water. The remaining photoresist and the Cr-layer were removed by acetone and MPM-E30, respectively.

## Fabrication of PDMS pad

We used a polydimethylsiloxane (PDMS) pad to flow culture medium and control the environmental conditions around the cells in the microchamber array. The PDMS pad was designed to have a square bubble-trap groove, which prevents interference with bright-field microscopic imaging by air bubbles in flowing media.

To create a mold for the bubble-trap groove, we spin-coated SU-8 3050 (Kayaku Advanced Materials) on a silicon wafer (ID 447, $\phi$ = 76.2 mm, University Wafer) and baked it at 95°C for 2 hr on a hot plate. The SU-8 layer was exposed to UV light on a mask aligner using a photomask and postbaked at 95°C for 2 hr. After cooled down to room temperature, the SU-8 photoresist was developed in the SU-8 developer (Kayaku Advanced Materials) and rinsed with isopropanol (Wako).

Part A and Part B of PDMS resin (SYLGARD 184 Silicone Elastomer Kit, DOW SILICONES) were mixed at 10:1 and poured onto the SU-8 mold. The air bubbles were removed under a decreased pressure for 30 min. The PDMS was cured at 65°C for 1 hour, and 20 mm × 20 mm square PDMS pad was cut out using a blade. We punched out two holes ($\phi$ = 2 mm) in the PDMS pad for the inlet and outlet, and 10-cm silicone tubes (SR-1554, Tigers Polymer Corp., outer $\phi$ = 2 mm, inner $\phi$ = 1 mm) were inserted into the holes. The tubes were fixed to the holes by gluing a small amount of PDMS around the tubes at the holes. This PDMS pad was washed in isopropanol by sonication and autoclaved for the single-cell measurements.

## Chemical decoration of coverslip and cellulose membrane

We washed the microfabricated coverslips by sonication in contaminon (Wako), ethanol (Wako), and 0.1 M NaOH solution (Wako). The washed coverslips were rinsed in milliQ water by sonication and dried at 140°C for 30 min. The washed coverslip was soaked in 1% (v/v) 3-(2-aminoethylaminopropyl) trimethoxysilane solution (Shinetsu Kagaku Kogyo) for 30 min and incubated at 140°C for 30 min to create an amino group on the glass surface. The treated coverslip was washed in milliQ water for 15 min and dried at 140°C for 30 min. 1 mg NHS-LC-LC-Biotin (Funakoshi) was dissolved in 25 µl dimethyl sulfoxide and dispersed in 1 ml phosphate buffer (0.1 mM, pH8.0). A total of 200 µl of this biotin solution was placed on the coverslip and incubated at room temperature for 4 hr. The biotin solution was removed by soaking the coverslip in milliQ water.

We prepared a streptavidin-decorated cellulose membrane to enclose cells in the microchamber array while retaining a flexible environmental control. First, a 3 cm × 3 cm square cellulose membrane (Spectra/Por7 Pre-treated RC Tubing MWCO:25kD) was cut out and washed in milliQ water for 10 min. The membrane was incubated in a 0.1 M NaIO$_4$ solution with gentle shaking for 4 hr at 25°C. After the wash in milliQ water, the treated membrane was incubated in a 500-µl solution of streptavidin hydrazide (Funakoshi) (10 µg/ml, dissolved in 0.1 mM phosphate buffer (pH7.0)) with gentle shaking for 14 hr at 25°C. The membrane was again washed in milliQ water and stored at 4°C.

## *E. coli* strains

We used two *E. coli* strains: MG1655 and MG1655 F3 *rpoS-mcherry* (MG1655 Δ*fliC*Δ*fimA*Δ*flu rpoS-mcherry*/pUA66-PrplS-*gfp*). MG1655 was used in the regrowth experiment from the stationary phases (*Figure 6*). MG1655 F3 *rpoS-mcherry* was used for analyzing the growth in constant environments (*Figures 5 and 7*). In MG1655 F3 *rpoS-mcherry*, the three genes, *fliC*, *fimA*, and *flu*, were deleted, and *mcherry* gene was inserted downstream of *rpoS* gene to express RpoS-mCherry translational fusion protein. This strain also expresses green fluorescent protein (GFP) from a low-copy plasmid, pUA66-PrplS-*gfp*, taken from a comprehensive library of fluorescent transcriptional reporters (*Zaslaver et al., 2006*).

## Culture conditions and sample preparation (exponential growth)

We used MG1655 F3 *rpoS-mcherry E. coli* strain and cultured the cells in M9 minimal medium (Difco) supplemented with 1/2 MEM amino acids solution (SIGMA) and 0.2% (w/v) glucose or glycerol as a carbon source. We set the cultivation temperature either at 37°C or 30°C.

To prepare *E. coli* cells for single-cell observation, we first inoculated a glycerol stock into a 3-ml culture medium and incubated it with shaking overnight under the same conditions of culture medium and temperature as those used in the time-lapse measurement. 30 µl of the overnight culture was inoculated in a 3-ml fresh medium and incubated with shaking until the optical density at $\lambda$ = 600 nm

reaches 0.1-0.3. This exponential-phase culture was diluted to $OD_{600}$ = 0.05, and 0.5 µl of the diluted cell suspension was spotted on the microchamber array on a biotin-decorated coverslip. A 5-mm × 5-mm streptavidin-decorated cellulose membrane was placed gently on the cell suspension on the coverslip, and an excess cell suspension was removed by a clean filter paper. A small piece of agar pad made with the culture medium and 1.5% (w/v) agar was placed on the cellulose membrane to maintain the culture conditions around the cells until tight streptavidin-biotin bonding was formed between the coverslip and the membrane. After 5-min incubation, the agar pad was removed, and the PDMS pad for medium perfusion was attached on the coverslip via a square-frame two-sided seal (Frame-Seal Incubation Chambers, Bio-rad). We immediately filled the device with the fresh medium and connected it to a syringe pump on the microscope stage.

## Culture conditions and sample preparation (regrowth from stationary phases)

We used *E. coli* MG1655 strain and cultured the cells in Luria-Bertani (LB) medium at 37°C. To prepare the cells for the time-lapse experiment, a glycerol stock of this strain was inoculated into a 2 ml LB medium and cultured with shaking for 15 hours. The cell culture was diluted in 50 ml fresh LB medium to $OD_{600}$ = 0.005 and again cultured with shaking as a pre-culture. For preparing the early-stationary-phase conditioned medium, 7 ml pre-culture cell suspension at 8 hr ($OD_{600} \approx 4.3$) was spun down at 2600 G for 12 min. The supernatant was filtered through a 0.22-µm filter. For preparing cells for time-lapse observation, a 10-µl pre-culture cell suspension at 8 hr was mixed with 240 µl early-stationary-phase conditioned medium. One µl of this diluted cell suspension was placed on the microchamber array on a biotin-decorated glass coverslip. A 5-mm × 5-mm streptavidin-decorated cellulose membrane was placed gently on the cell suspension on the coverslip, and an excess cell suspension was removed by a clean filter paper. A small piece of a conditioned medium agar pad made with 1.5% (w/v) agar was placed on a cellulose membrane to maintain the early stationary phase condition during the incubation. After 5-min incubation, the conditioned medium agar pad was removed, and the PDMS pad for medium perfusion was attached on the coverslip via a square-frame two-sided seal. We immediately filled the device with the conditioned medium and connected it to a syringe pump. We maintained the chamber filled with the conditioned medium until we started the time-lapse observation. The conditioned medium was flushed away immediately before starting the time-lapse measurement by flowing fresh LB medium. After flowing 2 ml fresh LB medium at 32 ml/hr, the flow rate was decreased and maintained at 2 ml/hr throughout the time-lapse measurement.

We followed the same procedures for the late stationary phase sample except that we sampled the cells and prepared the conditioned medium from a 24-hr pre-culture cell suspension ($OD_{600} \approx 3.0$).

## Time-lapse measurements and image analysis

We used Nikon Ti-E inverted microscope equipped with Plan Apo $\lambda$ 100× phase contrast objective (NA1.45), ORCA-R2 cooled CCD camera (Hamamatsu Photonics), Thermobox chamber (Tokai hit, TIZHB), and LED excitation light source (Thorlabs, DC2100). The microscope was controlled by Micromanager (*Edelstein et al., 2014*). In the exponential phase experiments, we monitored 25-30 microchambers in parallel in one measurement and acquired the phase-contrast, RpoS-mCherry fluorescence, and GFP fluorescence images from each position with a 3-min interval. We repeated the time-lapse measurement for each culture condition three times. In the regrowth experiment from the stationary phases, we monitored 150-250 microchambers in parallel with a 3-min interval and acquired only phase-contrast images.

We analyzed the time-lapse images by ImageJ (*Schneider et al., 2012*). We extracted the information of cell size (projected cell area), RpoS-mCherry fluorescence mean intensity, and GFP fluorescence mean intensity of individual cells along with division timings on each cell lineage for the exponential phase experiment. We extracted only division timings on each cellular lineage for the regrowth experiments from the stationary phases and used this information for further analysis.

## Data analysis
### Distributions and selection strength measures for division count

We calculated the distributions and selection strength measures of $D$ as follows. With the list of division counts $\{D\}$ for each lineage $\sigma$, the chronological and retrospective probabilities were evaluated

as $P_{\text{cl}}(\sigma) = 2^{-D(\sigma)}/N_0$ and $P_{\text{rs}}(\sigma) = 1/N_\tau$, respectively, where $N_0$ is the number of cells at $t = 0$ and $N_\tau$ is that at $t = \tau$. From these probabilities, the chronological and retrospective distributions of $D$ were obtained by summing the lineage probabilities for each division count, that is,

$$Q_{\text{cl}}(d) = \sum_{\sigma:D(\sigma)=d} P_{\text{cl}}(\sigma), \tag{21}$$

$$Q_{\text{rs}}(d) = \sum_{\sigma:D(\sigma)=d} P_{\text{rs}}(\sigma). \tag{22}$$

The selection strength measures, $S_{\text{KL}}^{(1)}[D]$ and $S_{\text{KL}}^{(2)}[D]$, were calculated as

$$S_{\text{KL}}^{(1)}[D] = \sum_{d \in D_{\text{supp}}} Q_{\text{cl}}(d) \ln \frac{Q_{\text{cl}}(d)}{Q_{\text{rs}}(d)}, \tag{23}$$

$$S_{\text{KL}}^{(2)}[D] = \sum_{d \in D_{\text{supp}}} Q_{\text{rs}}(d) \ln \frac{Q_{\text{rs}}(d)}{Q_{\text{cl}}(d)}, \tag{24}$$

where $D_{\text{supp}}$ is the support of both chronological and retrospective probabilities with respect to $D$, which is common between the two probabilities.

## Distributions and selection strength measures for time-averaged fluorescence intensity of RpoS-mCherry and GFP

We obtained the mean fluorescence intensity of RpoS-mCherry and GFP along with the genealogical trees in the time-lapse measurements of *E. coli* MG1655 F3 *rpoS-mcherry* strain. We analyzed the time-averaged fluorescence intensity of RpoS-mCherry and GFP as a lineage trait $X$ and evaluated their distributions, fitness landscapes, and selection strength measures (**Figure 7**). For each cell lineage, the time-averaged fluorescence intensity was calculated as

$$X(\sigma) = \frac{1}{N+1} \sum_{i=0}^{N} x_\sigma(t_i), \tag{25}$$

where $t_i = t_{\text{start}} + i\Delta t$ min ($t_{start}$ is the start time of the cell lineage; $\Delta t = 3$ min is the time-lapse interval), and $x_\sigma(t_i)$ is the mean fluorescence intensity at time $t_i$.

Generally, bin sizes for the fluorescence intensity affect the selection strength values. However, one can usually find the ranges of bin sizes where the results are relatively insensitive to the choice (**Nozoe et al., 2017**). Following an empirical rule, we set the bin width $\Delta X$ to

$$\Delta X = 0.4 * \text{IQR}(\{X\}), \tag{26}$$

where $\text{IQR}(X)$ is the interquartile range of the set of $X(\sigma)$ from all the cell lineages. Then, the interval was defined as $I_{x,\Delta X} = \left[x - \frac{\Delta X}{2}, x + \frac{\Delta X}{2}\right]$ for $x = \min(\{X\}), \min(\{X\}) + \Delta X, \cdots, \min(\{X\}) + (L-1)\Delta X$, where $L$ is the number of total bins given by $L = \lfloor \frac{\max(\{X\})-\min(\{X\})}{\Delta X} \rfloor + 2$.

We calculated the chronological and retrospective probability distributions of $X$ by

$$Q_{\text{cl}}(x) = \sum_{\sigma:X(\sigma)\in I_{x,\Delta X}} \frac{2^{-D(\sigma)}}{N_0}, \tag{27}$$

$$Q_{\text{rs}}(x) = \sum_{\sigma:X(\sigma)\in I_{x,\Delta X}} \frac{1}{N_\tau}. \tag{28}$$

$h(x)$ The fitness landscape was evaluated by

$$h(x) = \ln \frac{N_\tau}{N_0} \frac{Q_{\text{rs}}(x)}{Q_{\text{cl}}(x)}. \tag{29}$$

The selection strength measures were evaluated by

$$S_{\text{KL}}^{(1)}[X] = \sum_{l=0}^{L-1} Q_{\text{cl}}(\min(X) + l\Delta X) \ln \frac{Q_{\text{cl}}(\min(X)+l\Delta X)}{Q_{\text{rs}}(\min(X)+l\Delta X)}, \tag{30}$$

$$S_{\text{KL}}^{(2)}[X] = \sum_{l=0}^{L-1} Q_{\text{rs}}(\min(X) + l\Delta X) \ln \frac{Q_{\text{rs}}(\min(X)+l\Delta X)}{Q_{\text{cl}}(\min(X)+l\Delta X)}. \tag{31}$$

## Cumulant generating functions and cumulants

To plot the differential of the cumulant generating functions in **Figure 5F-H**, we evaluated

$$K'_D(\xi) = \frac{\sum\limits_{d \in D_{\text{supp}}} (d \ln 2) 2^{\xi d} Q_{\text{cl}}(d)}{\sum\limits_{d \in D_{\text{supp}}} 2^{\xi d} Q_{\text{cl}}(d)}$$ by changing $\xi$ from 0 to 1 with the step size 0.01.

We calculated the cumulative contributions of fitness cumulants to the population growth $W_n^{(X)}$ (**Figures 5A, 6E and 7F-H**) using a julia package, JuliaDiff/TaylorSeries.jl (**Benet and Sanders, 2019**; **Benet and Sanders, 2021**).

## Error estimations by resampling method

To evaluate the error ranges of the quantities calculated in the analysis, we created 20,000 randomly resampled datasets for each condition and reported the means and two standard deviation ranges in the results.

For the datasets of colony growth (*E. coli* and *M. smegmatis*), $N_\tau$ lineages were randomly sampled with replacement according to the probability weight $P_{\text{rs}}(\sigma)$ for each resampled dataset. In each resampled dataset, the initial number of cells was estimated as $\hat{N}_0 = \sum_{\sigma \in \{\sigma\}_{\text{resampled}}} 2^{-D(\sigma)}$.

For the datasets taken using the mother machines (*S. pombe* and L1210), we randomly sampled $N_0$ lineages with an equal weight, which corresponds to the chronological probability in this setting. $N_\tau$ was estimated as $\hat{N}_\tau = \sum_{\sigma \in \{\sigma\}_{\text{resampled}}} 2^{D(\sigma)}$.

## Simulating the effect of cell removal on population growth rates

We simulated cell population growth with cell removal using a custom C script. The gamma distributions were adopted as generation time distributions. We assigned the shape parameter to $k = 1, 2,$ or 5 and the scale parameter to $\theta = 2^{1/k} - 1$. The perturbation strength $\epsilon$ was changed from 0 to 0.2 with the interval 0.01.

As a pre-run, we started a simulation from a newborn cell and assigned its generation time randomly according to a pre-defined gamma probability distribution. We assumed that this cell divided into two daughter cells at the end of the generation. Each daughter cell was removed with probability $1 - 2^{-\epsilon}$ and assigned with generation time from the same pre-defined probability distribution if it escaped removal. Repeating this procedure, we let the population grow until all of the remaining cell lineages in the population exceed the maximum duration $T_{\text{max}} = 8.0$. The time to the next division of each cell lineage at $T_{\text{max}}$ was exported as the first division time in the main simulation. This pre-run was repeated 1000 cycles to export a sufficiently sizable list of first division times.

In the main simulation, we started from a progenitor cell with its division time randomly assigned from the first division time list exported in the pre-rum. For the daughter cells born from the first divisions and their descendants, the assignment of generation time and the cell removal were done as in the pre-run. We stopped further production of daughter cells in each lineage if it exceeded $T_{\text{max}} = 8.0$. We repeated this main simulation 1,000 cycles starting from different progenitor cells. The number of cell divisions in each cell lineage until $T_{\text{max}}$ was exported for analysis.

We calculated the population growth rate at each perturbation strength as

$$\Lambda(\epsilon) = \frac{1}{T_{\text{max}}} \ln \frac{N(T_{\text{max}}, \epsilon)}{1000}, \tag{32}$$

where $N(T_{\text{max}}, \epsilon)$ is the number of cell lineages at $T_{\text{max}}$ when the perturbation strength was $\epsilon$. The chronological and retrospective mean fitness of division count without cell removal was calculated as

$$\langle \tilde{h}(D) \rangle_{\text{cl}} = \sum_{\sigma=1}^{N(T_{\text{max}}, 0)} \frac{(D(\sigma) \ln 2) 2^{-D(\sigma)}}{1000}, \tag{33}$$

$$\langle \tilde{h}(D) \rangle_{\text{rs}} = \sum_{\sigma=1}^{N(T_{\text{max}}, 0)} \frac{D(\sigma) \ln 2}{N(T_{\text{max}}, 0)}. \tag{34}$$

When simulating the cell population with mother-daughter correlation time, we randomly assigned the generation time from the gamma probability distribution with its shape parameter $\frac{r\tau_m/\theta + k(1-r)}{1-r^2}$ and scale parameter $(1 - r^2)\theta$, where $\tau_m$ is the generation time of the mother cell, $r$ is the correlation coefficient of generation time between neighboring generations. The stationary distribution of this transition probability approximates the gamma distribution with shape parameter $k$ and scale

parameter $\theta$ to good precision with identical first and second-order moments irrespective of the parameters $k$, $\theta$, and $r$. In *Figure 4—figure supplement 1*, we fixed $k = 2$ and $\theta = \sqrt{2} - 1$ and set $r$ to 0, 0.2, 0.4, or 0.6.

## Data and code availability

The raw data obtained in this study, the Matlab codes for data analysis, and the C code for simulation have been deposited in Github repositories (https://github.com/Wakamoto-lab/LineageAnalysis, (copy archived at swh:1:rev:1865d167f1c24625c98d3c493a9a180b1aa2035d; *Yamauchi, 2021*), https://github.com/Wakamoto-lab/LineageAnalysis-Julia, (copy archived at swh:1:rev:e22fbce8a713582a18fbe2bcc57dc9078090f121; *Nozoe and Wakamoto, 2021*) and https://github.com/Wakamoto-lab/LineageSimulation (copy archived at swh:1:rev:ef1166620396835168ca9061851898993a091976; *Wakamoto, 2021*).

## Acknowledgements

We thank Tetsuya J Kobayashi and the members of the Wakamoto Lab for discussion. This work was supported by JST CREST Grant Number JPMJCR1927 (YW); JST ERATO Grant Number JPMJER1902 (YW); NIH Grant Number R01-GM097356 (EK); and Japan Society for the Promotion of Science KAKENHI Grant Number 17H06389 (YW), 19H03216 (YW), and 21K20672 (TN).

## Additional information

### Funding

| Funder | Grant reference number | Author |
| --- | --- | --- |
| Japan Science and Technology Agency | JPMJCR1927 | Yuichi Wakamoto |
| Japan Science and Technology Agency | JPMJER1902 | Yuichi Wakamoto |
| National Institute of General Medical Sciences | R01-GM097356 | Edo Kussell |
| Japan Society for the Promotion of Science | 17H06389 | Yuichi Wakamoto |
| Japan Society for the Promotion of Science | 19H03216 | Yuichi Wakamoto |
| Japan Society for the Promotion of Science | 21K20672 | Takashi Nozoe |

The funders had no role in study design, data collection and interpretation, or the decision to submit the work for publication.

### Author contributions

Shunpei Yamauchi, Conceptualization, Software, Formal analysis, Investigation, Visualization, Methodology, Writing - original draft, Writing - review and editing; Takashi Nozoe, Conceptualization, Software, Formal analysis, Validation, Investigation, Methodology, Writing - original draft, Writing - review and editing; Reiko Okura, Resources, Investigation; Edo Kussell, Conceptualization, Funding acquisition, Validation, Writing - original draft, Writing - review and editing; Yuichi Wakamoto, Conceptualization, Software, Supervision, Funding acquisition, Validation, Investigation, Visualization, Writing - original draft, Project administration, Writing - review and editing

### Author ORCIDs

Shunpei Yamauchi http://orcid.org/0000-0002-8530-4147
Takashi Nozoe http://orcid.org/0000-0003-2556-6484
Edo Kussell http://orcid.org/0000-0003-0590-4036
Yuichi Wakamoto http://orcid.org/0000-0002-6233-0844

Decision letter and Author response
Decision letter https://doi.org/10.7554/eLife.72299.sa1
Author response https://doi.org/10.7554/eLife.72299.sa2

## Additional files

### Supplementary files
• Transparent reporting form

### Data availability
All data generated or analyzed during this study and the Matlab codes for data analysis have been deposited in a GitHub repository (https://github.com/Wakamoto-lab/LineageAnalysis; copy archived at swh:1:rev:1865d167f1c24625c98d3c493a9a180b1aa2035d).

The following dataset was generated:

| Author(s) | Year | Dataset title | Dataset URL | Database and Identifier |
|---|---|---|---|---|
| Yamauchi S, Nozoe T, Okura R, Kussell E, Wakamoto Y | 2021 | LineageAnalysis | https://github.com/Wakamoto-lab/LineageAnalysis | Github, LineageAnalysis |

The following previously published datasets were used:

| Author(s) | Year | Dataset title | Dataset URL | Database and Identifier |
|---|---|---|---|---|
| Nozoe T, Kussell E, Wakamoto Y | 2018 | Data from: Inferring fitness landscapes and selection on phenotypic states from single-cell genealogical data | https://doi.org/10.5061/dryad.4539d | Dryad Digital Repository, 10.5061/dryad.4539d |
| Nakaoka H, Wakamoto Y | 2018 | Data from: Aging, mortality, and the fast growth trade-off of Schizosaccharomyces pombe | https://doi.org/10.5061/dryad.s2t5t | Dryad Digital Repository, 10.5061/dryad.s2t5t |
| Seita A, Nakaoka H, Okura R, Wakamoto Y | 2021 | Data from: Intrinsic growth heterogeneity of mouse leukemia cells underlies differential susceptibility to a growth-inhibiting anticancer drug | https://doi.org/10.5061/dryad.80gb5mkpr | Dryad Digital Repository, 10.5061/dryad.80gb5mkpr |

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

## Appendix 1

### Analytical calculations of fitness measures, selection strength, and the cumulants of a fitness landscape

To observe how the framework works, we show the exact form of $K_D(\xi)$ for a class of discrete probability distributions containing Poisson, binomial and negative binomial distributions. Let $\bar{D}$ and $\bar{D}\phi$ denote the mean and the variance of $Q_{cl}(D)$ respectively (i.e., $\phi$ is the Fano factor of division counts). When $Q_{cl}(D)$ is Poisson, binomial or negative binomial distributions, $\bar{D}$ and $\phi$ uniquely determine the form of probability distribution: $\phi = 1$ for Poisson; $\phi < 1$ for binomial; and $\phi > 1$ for negative binomial (*Appendix 1—figure 1A*). Then, $K_D(\xi)$ for these distributions have a closed form

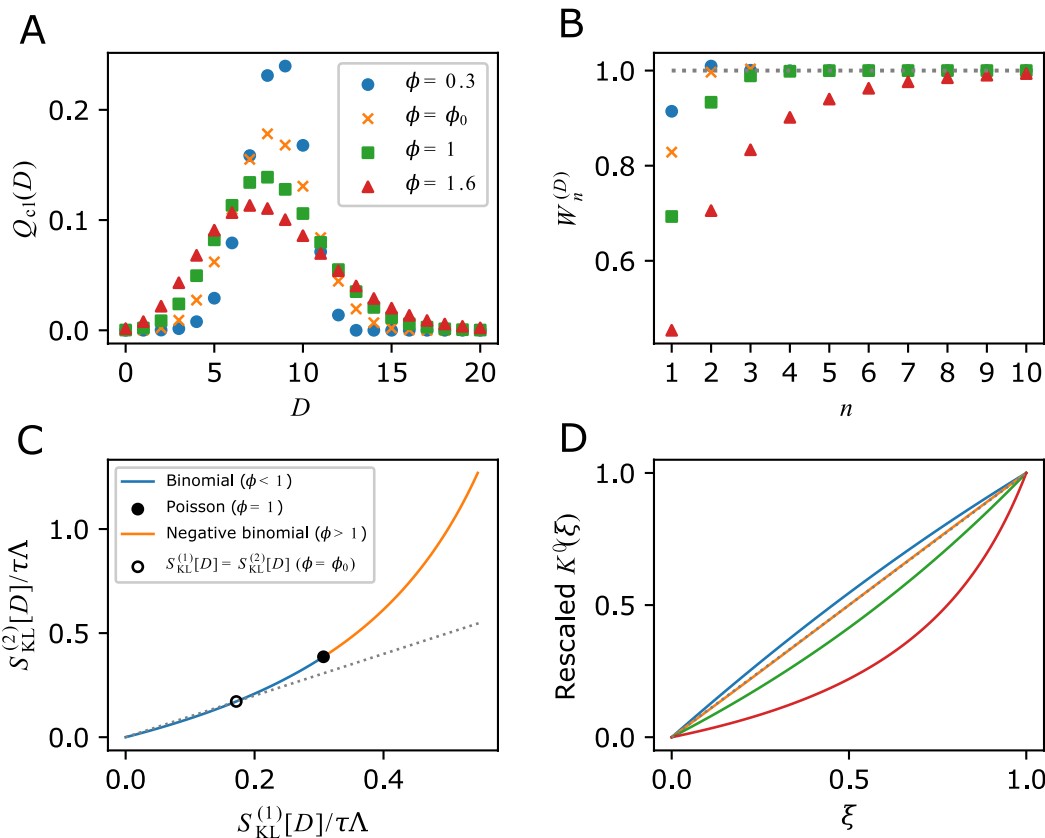

**Appendix 1—figure 1.** Analytical calculations of $K_D(\xi)$ and related relations given specific form of division count distributions. (**A**) Chronological division count distributions. $\phi = 0.3$ and $\phi = \phi_0 (= 0.5857...)$ are binomial, $\phi = 1$ is Poisson and $\phi = 1.6$ is negative binomial. $\bar{D} = 20(1 - \phi_0)$ is fixed. (**B**) Cumulative contributions of fitness cumulants. Parameter values are given in panel A legend. (**C**) The relation between two selection strength measures. Binomial (blue curve), Poisson (closed black circle) and negative binomial (orange curve) are indicated on the single curve plotted using *Equations 37 and 38* within the range of $0 < \phi < 2$. The point where $S_{KL}^{(1)}[D] = S_{KL}^{(2)}[D] (\phi = \phi_0)$ is indicated by the open black circle. The grey dotted line corresponds to $S_{KL}^{(1)}[D] = S_{KL}^{(2)}[D]$. (**D**) Convexity of $K_D'(\xi)$. Y-axis shows a rescaling of $K_D'(\xi)$ according to $\left(K_D'(\xi) - K_D'(0)\right) / \left(K_D'(1) - K_D'(0)\right)$. The same values of $\phi$ as in A are used; $\phi = 0.3$ (blue), $\phi = \phi_0$ (orange), $\phi = 1$ (green) and $\phi = 1.6$ (red). The grey dotted line indicates the case that $K_D'(\xi)$ is a linear function of $\xi$

$$
K_D(\xi) = \begin{cases} \bar{D}\dfrac{\ln\left(2^\xi (1-\phi) + \phi\right)}{1-\phi}, & \phi \neq 1 \\ \bar{D}\left(2^\xi - 1\right), & \phi = 1 \end{cases} \tag{35}
$$

(Appendix 3). We then immediately obtain

$$\tau\Lambda = K_D\left(1\right) = \begin{cases} \bar{D}\frac{\ln\left(2-\phi\right)}{1-\phi}, & \phi \neq 1 \\ \bar{D}, & \phi = 1 \end{cases} \tag{36}$$

Since $\lim_{\phi\to 2} K_D\left(1\right) = \infty$, $0 < \phi < 2$ is the range that the Fano factor of division counts can take within this scheme.

Using (**Equation 35**) allows us to calculate the cumulative contribution of cumulants of a fitness landscape $W_n^{(D)}$ (**Equation 14**). Plotting $W_n^{(D)}$ shows that the contribution of higher-order cumulants becomes significant when $\phi$ is large (**Appendix 1—figure 1B**). Also, evaluating the values of the derivative of **Equation 35** at $\xi = 0$ and $\xi = 1$, we have

$$\frac{S_{\mathrm{KL}}^{(1)}\left[D\right]}{\tau\Lambda} = 1 - \frac{K_D'\left(0\right)}{K_D\left(1\right)} = 1 - \frac{\left(1-\phi\right)\ln 2}{\ln\left(2-\phi\right)}, \tag{37}$$

$$\frac{S_{\mathrm{KL}}^{(2)}\left[D\right]}{\tau\Lambda} = \frac{K_D'\left(1\right)}{K_D\left(1\right)} - 1 = \frac{2\left(1-\phi\right)\ln 2}{\left(2-\phi\right)\ln\left(2-\phi\right)} - 1. \tag{38}$$

Therefore, $S_{\mathrm{KL}}^{(1)}\left[D\right]/\tau\Lambda$ and $S_{\mathrm{KL}}^{(2)}\left[D\right]/\tau\Lambda$ depend only on the Fano factor $\phi$. In particular, $S_{\mathrm{KL}}^{(1)}\left[D\right] = S_{\mathrm{KL}}^{(2)}\left[D\right]$ has 2 roots $\phi = 0, \phi_0(= 0.5857)$; $S_{\mathrm{KL}}^{(1)}\left[D\right] > S_{\mathrm{KL}}^{(2)}\left[D\right]$ if $0 < \phi < \phi_0$ and $S_{\mathrm{KL}}^{(1)}\left[D\right] < S_{\mathrm{KL}}^{(2)}\left[D\right]$ if $\phi_0 < \phi < 2$ (**Appendix 1—figure 1C**). Plotting $K_D'(\xi)$ confirms that the covexity direction changes around $\phi_0$ (**Appendix 1—figure 1D**). These analyses demonstrate how one can extract detailed information regarding selection in populations from $Q_{\mathrm{cl}}\left(D\right)$.

## Appendix 2

### Long-term limit for gamma-distributed uncorrelated generation times

To understand how inherent stochasticity affect long-term population growth rate and selection, we consider a cellular population in which cells divide stochastically following a probability distribution of generation times (interdivision times).

Let $g(x)$ and $z$ denote the probability density function of generation time $x$ and the mean number of offsprings per generation, respectively. We assume that the generation time correlation between parent and offspring can be ignored; i.e., $g(x)$ gives the probability density that offspring's generation time becomes $x$. The Malthusian parameter $\lambda$ is the real root of the so-called Euler-Lotka equation (**Fisher, 1930**):

$$z \int_0^\infty g(x) e^{-\lambda x} dx = 1. \tag{39}$$

We remark that (**Equation 39**) also holds for correlated generation times such as Markov models (**Lebowitz and Rubinow, 1974**) by reinterpreting $g(x)$ as the probability distribution of generation times of parent cells across a steadily growing population. In such cases, $g(x)$ depends on $z$, and we cannot treat $g(x)$ in (**Equation 43**) independent of $z = 2^\xi$. Here, we ignore any transgenerational correlations in generation time to illustrate the effect of the variation in generation time on $K_D(\xi)$ and selection strength measures with simple calculations. For this purpose, we further choose gamma distributions as $g(x)$, i.e.,

$$g(x) = \frac{x^{\alpha-1} e^{-x/\theta}}{\Gamma(\alpha) \theta^\alpha}, x \geq 0, \tag{40}$$

where $\alpha > 0$ is a shape parameter; and $\theta > 0$ is a scale parameter. In this case, the Malthusian parameter is

$$\lambda = \frac{z^{1/\alpha} - 1}{\theta}. \tag{41}$$

The probability distribution of division count $Q_{\mathrm{cl}}(D)$, in this case, is known as gamma count distribution (**Winkelmann, 1995**). Though any closed-form expression of the corresponding cumulant generating function is not known, it has a simple limiting form for $\tau \to \infty$ as shown below. We define the rescaled cumulant generating function by

$$\tilde{K}_D(\xi) := \lim_{\tau \to \infty} \frac{K_D(\xi)}{\tau}. \tag{42}$$

Since $\tilde{K}_D(\xi)$ represents the population growth rate, or Malthusian parameter with the mean number of offspring $z = 2^\xi$, we have

$$2^\xi \int_0^\infty g(x) e^{-\tilde{K}_D(\xi) x} dx = 1. \tag{43}$$

When $g$ is a gamma distribution with a shape parameter $\alpha$ and a scale parameter $\theta$, we obtain

$$\tilde{K}_D(\xi) = \frac{2^{\xi/\alpha} - 1}{\theta}, \tag{44}$$

and

$$\tilde{K}_D'(\xi) = \frac{2^{\xi/\alpha} \ln 2}{\alpha \theta}. \tag{45}$$

Note that $\alpha = 1$ corresponds to the case where division counts follow the Poisson distribution with mean $\theta^{-1}$. The scaled key quantities derived from $\tilde{K}_D(\xi)$ are as follows.

$$\Lambda = \tilde{K}_D(1) = \frac{2^{1/\alpha} - 1}{\theta}, \tag{46}$$

$$\tilde{K}_D'(0) = \frac{\ln 2}{\alpha \theta}, \tag{47}$$

$$\tilde{K}'_D(1) = \frac{2^{1/\alpha}\ln 2}{\alpha\theta} \,, \tag{48}$$

$$\tilde{S}^{(1)}_{\mathrm{KL}}[D] := \tilde{K}_D(1) - \tilde{K}'_D(0) \,, \tag{49}$$

and

$$\tilde{S}^{(2)}_{\mathrm{KL}}[D] := \tilde{K}'_D(1) - \tilde{K}_D(1). \tag{50}$$

Hence,

$$\frac{\tilde{S}^{(1)}_{\mathrm{KL}}[D]}{\Lambda} = 1 - \frac{\tilde{K}'_D(0)}{\tilde{K}_D(1)} = 1 - \frac{\ln 2}{\alpha\left(2^{1/\alpha}-1\right)} \,, \tag{51}$$

and

$$\frac{\tilde{S}^{(2)}_{\mathrm{KL}}[D]}{\Lambda} = \frac{\tilde{K}'_D(1)}{\tilde{K}_D(1)} - 1 = \frac{2^{1/\alpha}\ln 2}{\alpha\left(2^{1/\alpha}-1\right)} - 1 \,. \tag{52}$$

$\tilde{S}^{(2)}_{\mathrm{KL}}[D] > \tilde{S}^{(1)}_{\mathrm{KL}}[D]$ is always true for $0 < \alpha < \infty$ because

$$\begin{aligned} \frac{\tilde{S}^{(2)}_{\mathrm{KL}}[D] - \tilde{S}^{(1)}_{\mathrm{KL}}[D]}{\Lambda} &= \frac{(\gamma-2)e^\gamma + \gamma + 2}{e^\gamma - 1} \\ &> \frac{(\gamma-2)(\gamma+1) + \gamma + 2}{e^\gamma - 1} \\ &= \frac{\gamma^2}{e^\gamma - 1} > 0 \,, \end{aligned} \tag{53}$$

where $\gamma = \alpha^{-1}\ln 2$ and the inequality $e^\gamma > 1 + \gamma \ (\gamma > 0)$ are used.

Since the Taylor expansion of $\tilde{K}_D(\xi)$ at $\xi = 0$ is

$$\tilde{K}_D(\xi) = \frac{2^{\xi/\alpha}-1}{\theta} = \sum_{n\geq 1} \frac{\xi^n}{n!}\frac{1}{\theta}\left(\frac{\ln 2}{\alpha}\right)^n \,, \tag{54}$$

the time-scaled $n$-th order fitness cumulant is

$$\tilde{\kappa}_n := \lim_{\tau\to\infty}\frac{\kappa_n}{\tau} = \frac{1}{\theta}\left(\frac{\ln 2}{\alpha}\right)^n \,, n = 1, 2, \cdots \tag{55}$$

Therefore,

$$W_n = \frac{1}{\Lambda}\sum_{m=1}^{n}\frac{\tilde{\kappa}_m}{m!} = \frac{\sum_{m=1}^{n}\frac{1}{m!}\left(\frac{\ln 2}{\alpha}\right)^m}{2^{1/\alpha}-1}. \tag{56}$$

These results show that, unlike the central limit theorem, higher-order cumulants remain even in the long-term limit. Selection strength also remains in the long-term limit, which means that inherent stochasticity of generation times continuously introduces selection within a cellular population. Importantly, the time-scaled cumulants and the selection strength depend on $\alpha$. Therefore, the shape of generation time distributions influences the long-term population growth rate and selection. Since $\tilde{S}^{(2)}_{\mathrm{KL}}[D]$ is always greater than $\tilde{S}^{(1)}_{\mathrm{KL}}[D]$, the fitness variance is larger in the retrospective distribution than in the chronological distribution.

## Appendix 3

### Theoretical details on selection strength and cumulant generating function

#### The properties of the selection strength of division count

Below we derive several important properties of the selection strength of division count. We focus on the selection strength measure $S_{\mathrm{KL}}^{(1)}$ and write it as $S$ this section for conciseness. However, the conclusions are likewise valid for $S_{\mathrm{JF}}$ and $S_{\mathrm{KL}}^{(2)}$.

The most detailed description of cellular lineage statistics is based on individual lineages $\sigma$. From the definitions of $P_{\mathrm{cl}}(\sigma)$ and $P_{\mathrm{rs}}(\sigma)$ in the main text, the relation

$$P_{\mathrm{rs}}(\sigma) = P_{\mathrm{cl}}(\sigma)e^{D(\sigma)\ln 2 - \tau\Lambda} \tag{57}$$

is held. We define the selection strength of cellular lineages as

$$
\begin{aligned}
S[\sigma] &:= \sum_{\sigma} P_{\mathrm{cl}}(\sigma)\ln\frac{P_{\mathrm{cl}}(\sigma)}{P_{\mathrm{rs}}(\sigma)}\\
&= \sum_{\sigma} P_{\mathrm{cl}}(\sigma)\ln\frac{P_{\mathrm{cl}}(\sigma)}{P_{\mathrm{cl}}(\sigma)e^{D(\sigma)\ln 2 - \tau\Lambda}}\\
&= \tau\Lambda - \langle D(\sigma)\ln 2\rangle_{\mathrm{cl}},
\end{aligned}
\tag{58}
$$

where $\langle D(\sigma)\ln 2\rangle_{\mathrm{cl}} = \sum_{\sigma}(D(\sigma)\ln 2)P_{\mathrm{cl}}(\sigma)$

From the definition of fitness landscape (**Equation 1**),

$$
\begin{aligned}
\tilde{h}(d) &= \tau\Lambda + \ln\frac{Q_{\mathrm{rs}}(d)}{Q_{\mathrm{cl}}(d)}\\
&= \tau\Lambda + \ln\frac{\sum_{\sigma:D(\sigma)=d}P_{\mathrm{rs}}(\sigma)}{\sum_{\sigma:D(\sigma)=d}P_{\mathrm{cl}}(\sigma)}\\
&= \tau\Lambda + \ln\frac{\sum_{\sigma:D(\sigma)=d}P_{\mathrm{cl}}(\sigma)e^{D(\sigma)\ln 2 - \tau\Lambda}}{\sum_{\sigma:D(\sigma)=d}P_{\mathrm{cl}}(\sigma)}\\
&= d\ln 2.
\end{aligned}
\tag{59}
$$

On the other hand,

$$
\begin{aligned}
\langle D(\sigma)\ln 2\rangle_{\mathrm{cl}} &= \sum_{\sigma}(D(\sigma)\ln 2)P_{\mathrm{cl}}(\sigma) = \sum_{d}\sum_{\sigma:D(\sigma)=d}(D(\sigma)\ln 2)P_{\mathrm{cl}}(\sigma)\\
&= \sum_{d}(d\ln 2)\sum_{\sigma:D(\sigma)=d}P_{\mathrm{cl}}(\sigma) = \sum_{d}\tilde{h}(d)Q_{\mathrm{cl}}(d)\\
&= \langle\tilde{h}(D)\rangle_{\mathrm{cl}}.
\end{aligned}
\tag{60}
$$

This proves that the chronological mean fitness of cellular lineages equals the chronological mean fitness of division count.

Since $S[D] = \tau\Lambda - \langle\tilde{h}(D)\rangle_{\mathrm{cl}}$ and $S[\sigma] = \tau\Lambda - \langle D(\sigma)\ln 2\rangle_{\mathrm{cl}}$ (**Equations 3; 58**),

$$S[D] = S[\sigma] \tag{61}$$

is also held. This result shows that the selection strength of $D$ is equivalent to the selection strength of cellular lineages despite $D$ being a coarse-grained lineage trait.

Another important property of $S[D]$ is that it sets the maximum bound for the selection strength of any lineage traits. Now we consider the joint probability distributions of $D$ and lineage trait $X$, which we write $Q_{\mathrm{cl}}(d,x)$ and $Q_{\mathrm{rs}}(d,x)$. We define the joint selection strength as

$$S[D,X] := \sum_{d}\sum_{x}Q_{\mathrm{cl}}(d,x)\ln\frac{Q_{\mathrm{cl}}(d,x)}{Q_{\mathrm{rs}}(d,x)}. \tag{62}$$

Using $Q_{\mathrm{cl}}(d,x) = Q_{\mathrm{cl}}(d|x)Q_{\mathrm{cl}}(x)$ and $Q_{\mathrm{rs}}(d,x) = Q_{\mathrm{rs}}(d|x)Q_{\mathrm{rs}}(x)$,

$$
\begin{aligned}
S[D,X] &= \sum_{x}\left(\sum_{d}Q_{\mathrm{cl}}(d|x)\right)Q_{\mathrm{cl}}(x)\ln\frac{Q_{\mathrm{cl}}(x)}{Q_{\mathrm{rs}}(x)} + \sum_{d}\sum_{x}Q_{\mathrm{cl}}(d,x)\ln\frac{Q_{\mathrm{cl}}(d|x)}{Q_{\mathrm{rs}}(d|x)}\\
&= S[X] + S[D|X],
\end{aligned}
\tag{63}
$$

where $S[D|X] := \sum_{d} \sum_{x} Q_{\text{cl}}(d,x) \ln \frac{Q_{\text{cl}}(d|x)}{Q_{\text{rs}}(d|x)}$, and we used $\sum_{d} Q_{\text{cl}}(d|x) = 1$.

Likewise, $S[D,X]$ can also be decomposed as

$$S[D,X] = S[D] + S[X|D]. \tag{64}$$

However, $S[X|D] = 0$ because

$$
\begin{aligned}
h(d,x) :=\quad & \tau\Lambda + \ln \frac{Q_{\text{rs}}(d,x)}{Q_{\text{cl}}(d,x)} \\[6pt]
=\quad & \tau\Lambda + \ln \frac{\sum_{\sigma:D(\sigma)=d,\ X(\sigma)=x} P_{\text{rs}}(\sigma)}{\sum_{\sigma:D(\sigma)=d,\ X(\sigma)=x} P_{\text{cl}}(\sigma)} \\[6pt]
=\quad & d\ln 2 = \tilde{h}(d),
\end{aligned} \tag{65}
$$

and

$$
\begin{aligned}
S[X|D] :=\quad & \sum_{d} \sum_{x} Q_{\text{cl}}(d,x) \ln \frac{Q_{\text{cl}}(x|d)}{Q_{\text{rs}}(x|d)} \\[6pt]
=\quad & \sum_{d} \sum_{x} Q_{\text{cl}}(d,x) \ln \frac{Q_{\text{cl}}(d,x)Q_{\text{rs}}(d)}{Q_{\text{rs}}(d,x)Q_{\text{cl}}(d)} \\[6pt]
=\quad & \sum_{d} \sum_{x} Q_{\text{cl}}(d,x) \left\{ \tilde{h}(d) - h(d,x) \right\} = 0
\end{aligned} \tag{66}
$$

from (**Equation 1**) and (**Equation 65**). This leads to

$$S[D] = S[X] + S[D|X] \tag{67}$$

from (**Equation 63**) and (**Equation 64**). Furthermore, $S[D|X] \geq 0$ from Jensen's inequality. Thus,

$$S[D] \geq S[X]. \tag{68}$$

The equality is held when $D$ is a deterministic function of $X$. This inequality shows that $S[D]$ $(= S[\sigma])$ sets the maximum bound for the selection strength of any lineage trait $X$.

## The cumulant generating function $K_X(\xi)$ provides both chronological and retrospective fitness cumulants

In the main text, we introduced the cumulant generating function of $h(x)$ with respect to the chronological distribution $Q_{\text{cl}}(x)$,

$$K_X(\xi) := \ln\langle e^{\xi h(x)} \rangle_{\text{cl}} = \ln \sum_{x} e^{\xi h(x)} Q_{\text{cl}}(x). \tag{69}$$

This function can also be written as

$$K_X(\xi) = \sum_{n=1}^{\infty} \frac{\kappa_n^{(X)}}{n!} \xi^n \tag{70}$$

when the fitness cumulants $\kappa_n^{(X)}$ are all finite, and the Taylor expansion converges at $\xi$. Also,

$$\kappa_n^{(X)} = \left. \frac{d^n K_X(\xi)}{d\xi^n} \right|_{\xi=0}. \tag{71}$$

Below we prove that $K_X(\xi)$ also gives the fitness cumulants on the retrospective distributions.

We define a cumulant generating function on the retrospective probability as

$$R_X(\xi) := \ln\langle e^{\xi h(x)} \rangle_{\text{rs}} = \ln \sum_{x} e^{\xi h(x)} Q_{\text{rs}}(x). \tag{72}$$

This function can be expanded by the fitness cumulants of the retrospective statistics $\rho_n^{(X)}$ as

$$R_X(\xi) = \sum_{n=1}^{\infty} \frac{\rho_n^{(X)}}{n!} \xi^n. \tag{73}$$

Therefore,

$$\rho_n^{(X)} = \left. \frac{d^n R_X(\xi)}{d\xi^n} \right|_{\xi=0} . \tag{74}$$

For example, $\rho_1^{(X)} = \langle h(X) \rangle_{\mathrm{rs}}$ and $\rho_2^{(X)} = \mathrm{Var}[h(X)]_{\mathrm{rs}} = \langle h(X)^2 \rangle_{\mathrm{rs}} - \langle h(X) \rangle_{\mathrm{rs}}^2$.

Inserting $Q_{\mathrm{rs}}(x) = e^{h(x)-\tau\Lambda} Q_{\mathrm{cl}}(x)$ into (*Equation 72*),

$$\begin{aligned} R_X(\xi) &= \ln \sum_x e^{\xi h(x)} \left( e^{h(x)-\tau\Lambda} Q_{\mathrm{cl}}(x) \right) \\ &= -\tau\Lambda + \ln \sum_x e^{(\xi+1)h(x)} Q_{\mathrm{cl}}(x) \\ &= -\tau\Lambda + K_X(\xi + 1). \end{aligned} \tag{75}$$

Hence,

$$\frac{d^n R_X(\xi)}{d\xi^n} = \frac{d^n K_X(\xi+1)}{d\xi^n}, \tag{76}$$

for $n \geq 1$. This relation proves that evaluating $\frac{d^n K_X(\xi)}{d\xi^n}$ at $\xi = 1$ gives the $n$-th order fitness cumulant on the retrospective statistics; i.e.,

$$\rho_n^{(X)} = \left. \frac{d^n K_X(\xi)}{d\xi^n} \right|_{\xi=1} . \tag{77}$$

Furthermore, this leads to

$$\rho_n^{(X)} = \sum_{k=n}^{\infty} \frac{\kappa_k^{(X)}}{(k-n)!}, \tag{78}$$

from (*Equation 70*) and (*Equation 77*). Similarly, evaluating (*Equation 76*) at $\xi = -1$ gives

$$\kappa_n^{(X)} = \left. \frac{d^n R_X(\xi)}{d\xi^n} \right|_{\xi=-1} = \sum_{k=n}^{\infty} \frac{\rho_k^{(X)}(-1)^{k-n}}{(k-n)!} . \tag{79}$$

Analogously to (*Equation 12*), we can also expand the population growth rate in terms of the retrospective cumulants, by evaluating (*Equation 75*) at $\xi = -1$,

$$\tau\Lambda = K_X(0) - R_X(-1) = \sum_{n=1}^{\infty} \frac{(-1)^{n-1}\rho_n^{(X)}}{n!} . \tag{80}$$

For example, when the fitness distribution is Gaussian for the chronological statistics,

$$\begin{aligned} \langle h(X) \rangle_{\mathrm{rs}} &= \rho_1^{(X)} = \kappa_1^{(X)} + \kappa_2^{(X)} \\ &= \langle h(X) \rangle_{\mathrm{cl}} + \mathrm{Var}[h(X)]_{\mathrm{cl}}, \end{aligned} \tag{81}$$

$$\begin{aligned} \mathrm{Var}[h(X)]_{\mathrm{rs}} &= \rho_2^{(X)} = \kappa_2^{(X)} \\ &= \mathrm{Var}[h(X)]_{\mathrm{cl}}, \end{aligned} \tag{82}$$

since $\kappa_n^X = 0$ for $\forall n \geq 3$.

These results confirm that the function $K_X(\xi)$ contains the information of both chronological and retrospective statistics.

## Relationships between fitness cumulants and selection strength measures

In the main text, we have shown that the selection strength $S_{\mathrm{KL}}^{(1)}[X]$ corresponds to the contribution of the second or higher-order fitness cumulants to population growth, i.e.,

$$S_{\mathrm{KL}}^{(1)}[X] = \sum_{n=2}^{\infty} \frac{\kappa_n^{(X)}}{n!} . \tag{83}$$

or alternatively, by substituting (*Equation 79*) and (*Equation 80*) we obtain

$$
\begin{aligned}
S_{\mathrm{KL}}^{(1)}[X] &= \sum_{n=1}^{\infty} \frac{\rho_n^{(X)}(-1)^{n-1}}{n!} - \sum_{n=1}^{\infty} \frac{\rho_n^{(X)}(-1)^{n-1}}{(n-1)!} \\
&= \sum_{n=2}^{\infty} \frac{\rho_n^{(X)}(-1)^n}{n!}(n-1) .
\end{aligned}
\tag{84}
$$

Similar expressions can also be found for $S_{\mathrm{KL}}^{(2)}[X]$. Since $S_{\mathrm{KL}}^{(2)}[X] = \langle h(X) \rangle_{\mathrm{rs}} - \tau \Lambda$ (*Equation 4*), substituting (*Equation 80*) yields

$$
S_{\mathrm{KL}}^{(2)}[X] = \sum_{n=2}^{\infty} \frac{(-1)^n \rho_n^{(X)}}{n!} ,
\tag{85}
$$

or alternatively, by substituting (*Equation 78*) and (*Equation 12*) we obtain

$$
\begin{aligned}
S_{\mathrm{KL}}^{(2)}[X] &= \sum_{n=1}^{\infty} \frac{\kappa_n^{(X)}}{(n-1)!} - \sum_{n=1}^{\infty} \frac{\kappa_n^{(X)}}{n!} \\
&= \sum_{n=2}^{\infty} \frac{\kappa_n^{(X)}}{n!}(n-1) .
\end{aligned}
\tag{86}
$$

These show that both of $S_{\mathrm{KL}}^{(1)}[X]$ and $S_{\mathrm{KL}}^{(2)}[X]$ can be expanded by the chronological or retrospective fitness cumulants.

The difference between these two selection strength measures is

$$
S_{\mathrm{KL}}^{(2)}[X] - S_{\mathrm{KL}}^{(1)}[X] = \sum_{n=3}^{\infty} \frac{\kappa_n^{(X)}}{n!}(n-2) = \sum_{n=3}^{\infty} \frac{\rho_n^{(X)}(-1)^{n-1}}{n!}(n-2)
\tag{87}
$$

from (*Equation 83*) to (*Equation 86*). Thus, it depends only on the third or higher-order fitness cumulants.

Finally, another selection strength measure $S_{\mathrm{JF}}[X]$ can also be expanded by the fitness cumulants as

$$
S_{\mathrm{JF}}[X] = S_{\mathrm{KL}}^{(1)}[X] + S_{\mathrm{KL}}^{(2)}[X] = \sum_{n=2}^{\infty} \frac{\kappa_n^{(X)}}{(n-1)!} = \sum_{n=2}^{\infty} \frac{\rho_n^{(X)}(-1)^n}{(n-1)!}
\tag{88}
$$

from (*Equation 83*) to (*Equation 86*). When the chronological fitness distribution is Gaussian ($\kappa_n^{(X)} = 0$ for $\forall n \geq 3$),

$$
\begin{aligned}
S_{\mathrm{KL}}^{(1)}[X] &= S_{\mathrm{KL}}^{(2)}[X] = \frac{\kappa_2^{(X)}}{2} = \frac{\mathrm{Var}[h(X)]_{\mathrm{cl}}}{2}, \\
S_{\mathrm{JF}}[X] &= \kappa_2^{(X)} = \mathrm{Var}[h(X)]_{\mathrm{cl}}.
\end{aligned}
\tag{89}
$$

## Analytical calculations of $K_D(\xi)$ and related relations given specific form of division count distributions

Here we derive (*Equations 35–38*) in the main text. We begin with the case where $Q_{\mathrm{cl}}(D)$ follows a Poisson distribution. Let $\bar{D}$ denote the chronological mean division count.

$$
Q_{\mathrm{cl}}(D) = \frac{\bar{D}^D e^{-\bar{D}}}{D!}
\tag{90}
$$

By the definition of $K_D(\xi)$,

$$
K_D(\xi) = \ln \sum_{D \geq 0} 2^{\xi D} \frac{\bar{D}^D e^{-\bar{D}}}{D!} = \bar{D}\left(2^{\xi} - 1\right)
\tag{91}
$$

By the Taylor expansion of $2^{\xi} = e^{\xi \ln 2}$, the $n$-th order cumulant is $\kappa_n^{(D)} = \bar{D}(\ln 2)^n$. Since

$$
\tau \Lambda = K_D(1) = \bar{D},
\tag{92}
$$

we derive

$$W_n = \sum_{m=1}^{n} \frac{(\ln 2)^m}{m!}. \tag{93}$$

For example, $W_1^{(D)} = 0.693$, $W_2^{(D)} = 0.933$, $W_3^{(D)} = 0.988$, and $W_4^{(D)} = 0.998$. The first order derivative of $K_D(\xi)$ is

$$K_D'(\xi) = \bar{D}2^\xi \ln 2, \tag{94}$$

and thereby we have

$$\langle D \rangle_{\text{cl}} \ln 2 = K_D'(0) = \bar{D} \ln 2, \tag{95}$$

$$\langle D \rangle_{\text{rs}} \ln 2 = K_D'(1) = 2\bar{D} \ln 2, \tag{96}$$

$$\frac{S_{\text{KL}}^{(1)}}{\tau \Lambda} = 1 - \frac{K_D'(0)}{K_D(1)} = 1 - \ln 2 \simeq 0.31, \tag{97}$$

and

$$\frac{S_{\text{KL}}^{(2)}}{\tau \Lambda} = \frac{K_D'(1)}{K_D(1)} - 1 = 2 \ln 2 - 1 \simeq 0.39. \tag{98}$$

Next we derive $K_D(\xi)$ for binomial and negative binomial distributions. Let $\bar{D}$ and $\bar{D}\phi$ denote the mean and the variance of $Q_{\text{cl}}(D)$. When $Q_{\text{cl}}(D)$ is binomial,

$$Q_{\text{cl}}(D) = \binom{D_{\max}}{D} p^D (1-p)^{D_{\max}-D}, D = 0, 1, ..., D_{\max} \tag{99}$$

where $D_{\max}$ and $p$ satisfy $\bar{D} = D_{\max}p$ and $\bar{D}\phi = D_{\max}p(1-p)$; namely $D_{\max} = \bar{D}/(1-\phi)$ and $p = 1 - \phi$. Therefore,

$$\begin{aligned} K_D(\xi) &= \ln \sum_{D \geq 0} 2^{\xi D} \binom{D_{\max}}{D} p^D (1-p)^{D_{\max}-D} \\ &= D_{\max} \ln \left( 2^\xi p + 1 - p \right) \\ &= \frac{\bar{D}}{1-\phi} \ln \left( 2^\xi (1-\phi) + \phi \right). \end{aligned} \tag{100}$$

When $Q_{\text{cl}}(D)$ is negative binomial,

$$Q_{\text{cl}}(D) = \frac{\Gamma(\alpha+D)}{\Gamma(\alpha)D!} p^D (1-p)^\alpha, D = 0, 1, ... \tag{101}$$

where $\alpha$ and $p$ satisfy $\bar{D} = \alpha p/(1-p)$ and $\bar{D}\phi = \alpha p/(1-p)^2$; namely $\alpha = \bar{D}/(\phi-1)$ and $p = 1 - \phi^{-1}$. Therefore,

$$\begin{aligned} K_D(\xi) &= \ln \sum_{D \geq 0} 2^{\xi D} \frac{\Gamma(\alpha+D)}{\Gamma(\alpha)D!} p^D (1-p)^\alpha \\ &= -\alpha \ln \left( \frac{1-2^\xi p}{1-p} \right) \\ &= \frac{\bar{D}}{1-\phi} \ln \left( 2^\xi (1-\phi) + \phi \right). \end{aligned} \tag{102}$$

(*Equation 102*) is exactly the same as (*Equation 100*) as the function of $\bar{D}, \phi$, and $\xi$. In addition, (*Equation 91*) is the limiting form of (*Equation 100*) and (*Equation 102*) as $\phi \to 1$. Thus, (*Equation 35*) in the main text represents $K_D(\xi)$ for Poisson, binomial or negative binomial $Q_{\text{cl}}(D)$.

The Taylor expansion of (*Equation 35*) is obtained as follows:

$$\begin{aligned} \frac{K_D(\xi)}{\bar{D}} &= \sum_{m \geq 1} \frac{(\phi-1)^{m-1}}{m} \left( 2^\xi - 1 \right)^m \\ &= \sum_{m \geq 1} \frac{(\phi-1)^{m-1}}{m} \sum_{k \geq 0} \binom{m}{k} (-1)^{m-k} \sum_{n \geq 0} \frac{(k\xi \ln 2)^n}{n!} \\ &= \sum_{n \geq 0} \frac{(\xi \ln 2)^n}{n!} c_n(\phi), \end{aligned} \tag{103}$$

where

$$c_n\left(\phi\right) = \sum_{m \geq 1} \frac{(\phi-1)^{m-1}}{m} \sum_{k=0}^{m} k^n \binom{m}{k} (-1)^{m-k}.$$ (104)

For the first five terms, for example, we have

$$c_1\left(\phi\right) \quad = 1$$ (105a)

$$c_2\left(\phi\right) \quad = \phi$$ (105b)

$$c_3\left(\phi\right) \quad = \phi(2\phi - 1)$$ (105c)

$$c_4\left(\phi\right) \quad = \phi(6\phi^2 - 6\phi + 1)$$ (105d)

$$c_5\left(\phi\right) \quad = \phi(24\phi^3 - 36\phi^2 + 14\phi - 1)$$ (105e)

$\kappa_n^{(D)} = \bar{D}c_n(\phi)(\ln 2)^n$ gives the $n$-th order cumulant.

The first order derivative of (*Equation 35*) is

$$K_D'\left(\xi\right) = \frac{2^\xi \bar{D} \ln 2}{2^\xi (1-\phi)+\phi},$$ (106)

and thereby we obtain

$$\langle D \rangle_{\text{cl}} \ln 2 = K_D'\left(0\right) = \bar{D} \ln 2,$$ (107)

$$\langle D \rangle_{\text{rs}} \ln 2 = K_D'\left(1\right) = \frac{2\bar{D} \ln 2}{2 - \phi},$$ (108)

$$\frac{S_{\text{KL}}^{(1)}[D]}{\tau \Lambda} = 1 - \frac{K_D'(0)}{K_D(1)} = 1 - \frac{(1-\phi)\ln 2}{\ln(2-\phi)},$$ (109)

and

$$\frac{S_{\text{KL}}^{(2)}[D]}{\tau \Lambda} = \frac{K_D'(1)}{K_D(1)} - 1 = \frac{2(1-\phi)\ln 2}{(2-\phi)\ln(2-\phi)} - 1.$$ (110)

(*Equation 109*) and (*Equation 110*) equal if and only if

$$\frac{(2-\phi)\ln(2-\phi)}{(4-\phi)(1-\phi)} = \frac{\ln 2}{2}$$ (111)

This equation has two roots $\phi = 0$ and $\phi = \phi_0 = 0.5857...$ and LHS >RHS if and only if $0 < \phi < \phi_0$.

