## [Editor Report]

This manuscript presents a general statistical framework to infer selection on a quantitative trait, based on measurements of the values of this trait along related cell lineages. The manuscript provides both a detailed explanation of the mathematical underpinnings of the method and an illustration of its application to existing and new cell lineage datasets. This is a general framework and is not tailored to particular growth models or environmental conditions, making it applicable to broad examples of exponentially growing populations.

---

## [Decision Letter]

**Decision letter after peer review:**

Thank you for submitting your article "A unified framework for measuring selection on cellular lineages and traits" for consideration by *eLife*. Your article has been reviewed by 3 peer reviewers, and the evaluation has been overseen by a Reviewing Editor and Aleksandra Walczak as the Senior Editor. The following individual involved in review of your submission has agreed to reveal their identity: Srividya Iyer-Biswas (Reviewer #3).

Essential revisions:

1. The authors should restructure the paper so it is accessible to the broad audience of *eLife*. The main issues to keep in mind:

i. Better clarify of the biological questions that the approach can address and the biological insights that it can provide. Tangible connections to specific biological systems would strength the work.

ii. Better structure the results and present the main points in an accessible language to those not interested in the mathematical intricacies.

2. Better clarify the definitions and measures used for fitness, selection, evolutionary advantages, etc in the paper and contrast them with the common notion of fitness as the instantaneous growth. See the comments regarding the definitions of selection strength by reviewers #1 and #2.

3. Discuss the impact of inherent stochasticity in division events on the results and the outcome of lineage evolution.

4. Discuss how the mathematical formalism and results acquired for the stationary state can be applied (or modified) to address non-stationary conditions that are more relevant for the processes discussed in the manuscript. See comment 3 by reviewer #3.

The reviews below contain a number of other suggestions that we encourage you to consider.

*Reviewer #1 (Recommendations for the authors):*

– In the Introduction it is mentioned that « cell population's growth rate becomes greater than the mean division rate ». Can the formalism presented here describe this in a simple way?

– It could be useful for a broader audience to explicitly explain how skewness and cumulants are related.

– It could be good to define W1 and W2 in the theoretical background section.

– Figure 7: I recommend to explicitly tell what x is.

– How do definitions depend on the discrete or continuous nature of x? Practically, do we need to, say in Figure 7, bin x to compute different functions? How is this done?

*Reviewer #2 (Recommendations for the authors):*

The main concepts were previously proposed and published by the same authors (ref 13). The new clarification and applications are welcome but the scope of their novelty and impact is not totally obvious. Currently, the paper tends to read as a technical paper with new observations that are intriguing but not totally understood (e.g. the difference between stationary and non-stationary growth). It would benefit from a clarification of the biological questions that the approach can address and the biological insights that it can provide.

Selection is most commonly thought to act on traits and, in constant environments, quantified by the instantaneous growth. This quantity is often identified to fitness with a straightforward relation to adaptation at the population level. The reduction of fitness variance that the authors mention (in the abstract and line 260) is derived in this context. The paper takes a different perspective and it would be helpful to contrast it more clearly to this more usual approach: Why and when is it justified to define selection at the level of lineage trait? Why should we be interested in multiple definitions of selection strength in this context? What can we expect to learn?

Further explaining the questions that the formalism intend to address is needed or the paper may appear as a formal exercise to solve a problem that the authors artificially created, i.e. introducing multiple measures of selection strength on an unusual quantity. Further explaining the biological insights that the framework can provide is also needed if the intent is to reach readers interested in applications and not only mathematical technicalities.

Some questions along those lines:

Why multiple definitions of selection strength? Is it just a matter of quantifying the difference between Q_{rs}(x) and Q_{cl}(x) which cannot be reduced to a scalar quantity? What information is in principle contained in the difference? What is the biological interpretation of the observation that it can be reduced to 2 numbers in many cases (when higher order cumulant are negligible)?

In applications, it appears that interpretable conclusions are mainly drawn from two quantities: S_KL(D) for global selection and S_rel(X) for selection of a specific trait. In the current understanding of the approach, are these the quantities that one should compute to reach biological insights in practical applications?

Can we use the approach to rule out that a trait is under selection? If so, what would be the statistical evidence?

How critical is the formalism: can the authors derive a biological conclusion that would not be accessible without it?

The application of the method to non time-invariant conditions (regrowth, changing environments) is not completely clear. The results should depend on the time-window and important information pertaining to selection should be contained in the time evolution. The observation that empirical observation that S_KL^2^ differs from S_KL^1^ in this context is intriguing but its origins and implications unclear.

The authors stress that their model is independent from mechanisms, which makes it broadly applicable. But only correlations can be assessed which may limit the identification what drives selection.

What is the relationship to adaptation and evolution? The abstract raises the question but no further mention of adaptation is made in the rest of the paper.

Heredity is generally as important as selection: is it within or beyond the scope of the framework?

*Reviewer #3 (Recommendations for the authors):*

I recommend addressing the specific concerns raised through appropriate discussions and clarifications in the manuscript text.

---

## [Author Response]

Essential revisions:1. The authors should restructure the paper so it is accessible to the broad audience of eLife. The main issues to keep in mind:i. Better clarify of the biological questions that the approach can address and the biological insights that it can provide. Tangible connections to specific biological systems would strength the work.ii. Better structure the results and present the main points in an accessible language to those not interested in the mathematical intricacies.

We thank the reviewers for the suggestion. To address this issue, we restructured the manuscript significantly as follows.

First, we added the section titled "Examples of biological questions" immediately after the Introduction. We present three examples of fundamental biological questions for which this framework of cell lineage statistics is indispensable. This section also clarifies which quantities one needs to evaluate from experimental data to gain insights into each problem.

Second, we significantly reduced the mathematical descriptions in the main text and limited them only to the essential equations required for understanding the meanings of the quantities and interpreting the experimental results. We also moved most of the contents for model applications to Appendices to minimize the usage of equations and to present only the essential conclusions in the main text.

Third and lastly, we added more intuitive and plain explanations of the quantities where we first introduced them in the main text. Furthermore, in each application to the experimental data, we explained the biological insights gained by the results in more detail.

We hope these modifications and clarifications have made this manuscript more accessible to the broad audience of *eLife*. Please also see our replies to reviewers #1 and #2.

2. Better clarify the definitions and measures used for fitness, selection, evolutionary advantages, etc in the paper and contrast them with the common notion of fitness as the instantaneous growth. See the comments regarding the definitions of selection strength by reviewers #1 and #2.

We now include a glossary of the terms as Box 1 alongside a figure and explain the terms of fitness, fitness landscape, selection, selection strength, and cumulants. We also explain their standard usage in evolutionary biology and clarify their similarities and differences in this framework.

In addition, we clarify when the lineage-based consideration of fitness becomes indispensable in the new section, “Examples of biological questions,” referring to specific biological questions (L77-89).

The shared and distinct properties of the different selection strength measures introduced in this study are also clarified in the revised manuscript (L136-142 and Results). Please see our replies to the specific comments by reviewers #1 and #2.

3. Discuss the impact of inherent stochasticity in division events on the results and the outcome of lineage evolution.

We now clarify the impact of inherent stochasticity in interdivision time (generation time) on overall selection strength (SKL(1)[D]) and relative selection strength for a lineage trait (Srel[X]) in the Discussion (L519-526). We also explain the impact of stochasticity on the long-term growth rate of the population and selection (L327-334), referring to the results from the analytical model in Appendix 2. We demonstrate that the statistical properties of stochasticity influence both growth and selection even in the long-term limit. Please also see our replies to reviewers #1 and #3.

4. Discuss how the mathematical formalism and results acquired for the stationary state can be applied (or modified) to address non-stationary conditions that are more relevant for the processes discussed in the manuscript. See comment 3 by reviewer #3.

We now explicitly state in the text that this formalism can be applied to non-stationary conditions without modifications because evaluating fitness and selection requires only the information of division counts and trait dynamics in cell lineages (L62-65 and L492-493). However, we also clarify a limitation of this framework from this evaluation scheme: it cannot report any potential influences from uncharacterized factors, such as heterogeneous environments around cells and non-quantified traits (L493-496).

We also clarify the importance of the time windows for the results, especially when applied to non-stationary conditions (L536-539). Please see our replies to the comments by reviewers #1 and #3.

The reviews below contain a number of other suggestions that we encourage you to consider.Reviewer #1 (Recommendations for the authors):– In the Introduction it is mentioned that « cell population's growth rate becomes greater than the mean division rate ». Can the formalism presented here describe this in a simple way?

Yes, Equation 8 with *X* = *D* describes this effect. We now explain explicitly that the selection strength measure SKL(1)[D] corresponds to growth rate gain from growth (fitness) heterogeneity (L243-244).

– It could be useful for a broader audience to explicitly explain how skewness and cumulants are related.

We appreciate your suggestion. We now explain "cumulants" in Box 1 and clarify what first, second, and third-order cumulants represent. We also explain how the third-order cumulant is related to the skewness of a distribution.

– It could be good to define W1 and W2 in the theoretical background section.

We again appreciate your suggestion. The definitions of *W_1_* and *W_2_* become clear only after introducing the cumulant expansion. We therefore still believe it is more appropriate to define *W_1_* and *W_2_* in the Results section. Instead of showing the definitions in the Theoretical background section, we explicitly wrote the relations of *W_1_* and *W_2_* to ⟨h(X)⟩cl and Var[h(X)]cl in Results after the definition of *W_n_* (L262-263).

– Figure 7: I recommend to explicitly tell what x is.

We clarified in the legend of Figure 7A and B which quantities were adopted as lineage traits and what *x* represents.

– How do definitions depend on the discrete or continuous nature of x? Practically, do we need to, say in Figure 7, bin x to compute different functions? How is this done?

Thank you for pointing out an important practical issue. If *x* is a trait that takes continuous values, we need to bin the values of *x* for calculating fitness landscapes and selection strength from experimental data. Bin width does affect the results, but we can usually find a range of bin width in which the results are relatively insensitive to the choice (see ref. 13). In this study, we set the bin widths for the time-averaged RpoS-mCherry and GFP fluorescence intensities as 0.4 x (interquartile ranges) of the data from all the conditions following the rule that empirically works. We added explanations on this to Materials and methods (L795-796).

Reviewer #2 (Recommendations for the authors):The main concepts were previously proposed and published by the same authors (ref 13). The new clarification and applications are welcome but the scope of their novelty and impact is not totally obvious. Currently, the paper tends to read as a technical paper with new observations that are intriguing but not totally understood (e.g. the difference between stationary and non-stationary growth). It would benefit from a clarification of the biological questions that the approach can address and the biological insights that it can provide.

We appreciate the thoughtful advice. To clarify new biological questions that this approach can address, we added a section titled "Examples of biological questions" immediately after the Introduction. This new section presents three examples of fundamental biological questions and explains why this cell lineage analysis framework is indispensable.

In addition, we also mention new insights gained by the analyses in the experimental Results sections. Although gaining a more profound understanding of the results still requires further investigations, we believe the added information would clarify the types of biological questions for which this framework becomes valuable.

Selection is most commonly thought to act on traits and, in constant environments, quantified by the instantaneous growth. This quantity is often identified to fitness with a straightforward relation to adaptation at the population level. The reduction of fitness variance that the authors mention (in the abstract and line 260) is derived in this context. The paper takes a different perspective and it would be helpful to contrast it more clearly to this more usual approach: Why and when is it justified to define selection at the level of lineage trait? Why should we be interested in multiple definitions of selection strength in this context? What can we expect to learn?

We again thank the reviewer for an insightful suggestion. We now explain when a lineage-based analysis of traits and selection becomes indispensable in the "Examples of biological questions" section (L77-89). We also added discussions on reduction of fitness variance, clarifying the differences and similarities of the contexts (L540-548). Please also see our reply to the comment 4 of reviewer #1.

The lineage-based analysis is required especially when growth and traits fluctuate rapidly over time and when the traits affect growth with delays. Under these conditions, instantaneous correlations between traits and growth might not report their relations correctly. On the other hand, the cell lineage-based analysis of this framework can take the whole dynamics of traits in cell lineages into account. For example, if we expect that absolute expression levels are essential for fitness, the expression level averaged in each cell lineage can be employed as the lineage trait. Furthermore, when large fluctuations are expected to affect cell fates and promote or suppress growth (ref. 24, for example), variances of expression levels along the cell lineages can be taken as lineage traits. Therefore, assuming a cell lineage as a unit of selection can significantly extend the choice of traits, including time-dependent properties.

We also appreciate your question on why we need multiple selection strength measures. As we now explain explicitly in the main text, these measures share a similar property of reporting the overall correlations between traits and fitness (L136-137). However, they also have critical differences regarding additional selection effects they represent: SKL(1) for growth rate gain, SKL(2) for additional loss of growth rate under perturbations, and their difference SKL(2)−SKL(1) for the effect of selection on fitness variance. We restructured the sections in Results and clarified these important meanings of the different selection strength measures.

Further explaining the questions that the formalism intend to address is needed or the paper may appear as a formal exercise to solve a problem that the authors artificially created, i.e. introducing multiple measures of selection strength on an unusual quantity. Further explaining the biological insights that the framework can provide is also needed if the intent is to reach readers interested in applications and not only mathematical technicalities.

As we stated in our reply to the comment 1, we included a new section titled "Examples of biological questions" to clarify the biological questions we intended to address using this framework. We also explained in more detail what insights we gained from the experimental data analyses.

Some questions along those lines:Why multiple definitions of selection strength? Is it just a matter of quantifying the difference between Q_{rs}(x) and Q_{cl}(x) which cannot be reduced to a scalar quantity? What information is in principle contained in the difference? What is the biological interpretation of the observation that it can be reduced to 2 numbers in many cases (when higher order cumulant are negligible)?

As we stated in our reply to the comment 2, the selection measures SKL(1) and SKL(2) represent distinct selection effects (growth rate gain or additional loss of growth rate under perturbations) on cellular populations. Furthermore, the difference of the selection strength measures, SKL(2)−SKL(1), represents the effect of selection on fitness variances.

The situations where the contributions of higher-order cumulants are significant indicate that the fitness distributions are far from Gaussian due to significant skew or multiple peaks. Therefore, the higher-order cumulants can suggest the existence of sub-populations in the cellular populations. We now explicitly explain this role of higher-order cumulants in the Results and Discussion (L291-294 and L508-514).

In applications, it appears that interpretable conclusions are mainly drawn from two quantities: S_KL(D) for global selection and S_rel(X) for selection of a specific trait. In the current understanding of the approach, are these the quantities that one should compute to reach biological insights in practical applications?

We now clarify what quantities we should compute depending on the types of biological questions in the new section, "Examples of biological questions." When we aim to know how strong the overall selection is in the population and how much growth rate gain is obtained from fitness heterogeneity, one needs to quantify SKL(1)[D]. On the other hand, if we need to know how state differences of a trait of interest correlate with fitness and selection, one should compute the chronological distribution Qcl(x), the fitness landscape h(x), and the relative selection strength Srel[X], as those quantities have different meanings. For example, when we find a significant change in Srel[X], two extreme scenarios can be considered: (a) h(x) is unchanged, but Qcl(x) changes; (b) the distribution Qcl(x) is unchanged, but the fitness landscape h(x) changes. In reality, h(x) and Qcl(x) can change simultaneously depending on the traits and conditions. Therefore, evaluating all of these quantities is essential for understanding the underlying biological processes. We now explain this important point, referring to the experimental results (L458-464).

Can we use the approach to rule out that a trait is under selection? If so, what would be the statistical evidence?

Yes, a trait whose selection strength Srel[X] is zero is not under selection in a population. However, practically speaking, computing Srel[X] for any empirical lineage data will always find a positive value for this quantity. Therefore, it is helpful to compare the measured Srel[X] with those computed from the lineage data for which their correspondences between division counts and trait values are randomized (L427-430). If the trait is under selection, the measured Srel[X] will become more significant than the randomized values.

We now include a figure showing the levels of Srel[X] of randomized data for the experiments in which we evaluated the RpoS-mCherry and GFP expression levels under different culture conditions (Figure 7 – supplement 1).

How critical is the formalism: can the authors derive a biological conclusion that would not be accessible without it?

Related to our reply to comment 2, this formalism is critical if one needs to evaluate fitness and selection for traits with significant temporal fluctuations or delayed effects on fitness. Furthermore, this cell lineage-based framework is indispensable if we consider traits that can be defined only for cell lineages, not for individuals, such as the "variableness" of expression levels. Consequently, this framework significantly extends the choice of traits, including time-dependent properties.

We now explain this important point more clearly early in the manuscript in the section, "Examples of biological questions" (L77-89).

The application of the method to non time-invariant conditions (regrowth, changing environments) is not completely clear. The results should depend on the time-window and important information pertaining to selection should be contained in the time evolution. The observation that empirical observation that S_KL^2^ differs from S_KL^1^ in this context is intriguing but its origins and implications unclear.

We appreciate another insightful comment. It is correct that the time window affects the selection strength values. The purpose of this analysis on the regrowth of *E. coli* was to examine whether there was a significant difference in the selection strength depending on how long cell populations were placed under stationary phase conditions. For this purpose, it is appropriate to compare the results between the conditions (regrowth from the early stationary phase or the late stationary phase), fixing the time window for regrowth. Therefore, the results are valid at least in this time scale, but clarification of the selection in the longer time scales requires a more detailed characterization of lag time distributions under both conditions.

The significantly larger SKL(2) than SKL(1) indicates a strong positive skew of the division count distribution and an existence of distinct subpopulations. Although these conclusions are already clear from the lineage trees (Figure 7D) and the division count distribution (Figure 7F), the selection strength measures permit quantitative evaluation and comparison of the differences in a unified manner.

We added sentences to Discussion to explain that our conclusions from the analyses on regrowth depends on the time window (L536-539). In addition, we explained the implications of the larger SKL(2) than SKL(1) (L603-611).

The authors stress that their model is independent from mechanisms, which makes it broadly applicable. But only correlations can be assessed which may limit the identification what drives selection.

It is true that our approach can assess only correlations between traits and fitness. Revealing causal traits requires additional experiments and analyses in each phenomenon. Nevertheless, we believe this framework is valuable for narrowing down the candidates of traits whose heterogeneity influences fitness and selection; such causal traits should have large selection strength. We now discuss these limitations and utilities of this framework in Discussion (L496-499).

What is the relationship to adaptation and evolution? The abstract raises the question but no further mention of adaptation is made in the rest of the paper.

We appreciate the reviewer for pointing out the lack of explanations on the linkage to adaptation. We added a paragraph to the Discussion in which we discuss the implications of the results from the experimental data analyses for adaptation and evolution. We remark on the importance of growth heterogeneity structures within cellular populations for adaptation and discuss the advantage of this framework in characterizing such structures (L503-514).

Heredity is generally as important as selection: is it within or beyond the scope of the framework?

Again, we thank the reviewer for an insightful comment. We agree that heredity is also important for growth and evolution of a population. A straightforward application of this framework to this problem is to take correlation length (e.g., a half-life of autocorrelation of a particular trait) along a cell lineage as a lineage trait *X*. We now discuss the possibility of such an application, citing a reference that revealed modes of heredity could also be the target of the natural selection (L515-518).

Reviewer #3 (Recommendations for the authors):I recommend addressing the specific concerns raised through appropriate discussions and clarifications in the manuscript text.

We thank the reviewer for the constructive suggestion. As we replied above, we addressed the specific concerns by adding the discussions and clarifications to the text.